EMBO
Molecular Medicine

# Electron transport chain capacity expands yellow fever vaccine immunogenicity

Darren ZL Mok [ID][1], Danny JH Tng [ID][1,2], Jia Xin Yee[1,3], Valerie SY Chew[1,3], Christine YL Tham[1,3], Justin SG Ooi [ID][1], Hwee Cheng Tan[1], Summer L Zhang[1], Lowell Z Lin [ID][1], Wy Ching Ng [ID][1], Lavanya Lakshmi Jeeva[4], Ramya Murugayee[4], Kelvin K-K Goh[5], Tze-Peng Lim[5], Liang Cui[6], Yin Bun Cheung[7,8], Eugenia Z Ong[1,3], Kuan Rong Chan[1], Eng Eong Ooi [ID][1,3,9,10 ✉] & Jenny G Low [ID][1,2,3 ✉]

## Abstract

Vaccination has successfully controlled several infectious diseases although better vaccines remain desirable. Host response to vaccination studies have identified correlates of vaccine immunogenicity that could be useful to guide development and selection of future vaccines. However, it remains unclear whether these findings represent mere statistical correlations or reflect functional associations with vaccine immunogenicity. Functional associations, rather than statistical correlates, would offer mechanistic insights into vaccine-induced adaptive immunity. Through a human experimental study to test the immunomodulatory properties of metformin, an anti-diabetic drug, we chanced upon a functional determinant of neutralizing antibodies. Although vaccine viremia is a known correlate of antibody response, we found that in healthy volunteers with no detectable or low yellow fever 17D viremia, metformin-treated volunteers elicited higher neutralizing antibody titers than placebo-treated volunteers. Transcriptional and metabolomic analyses collectively showed that a brief course of metformin, started 3 days prior to YF17D vaccination and stopped at 3 days after vaccination, expanded oxidative phosphorylation and protein translation capacities. These increased capacities directly correlated with YF17D neutralizing antibody titers, with reduced reactive oxygen species response compared to placebo-treated volunteers. Our findings thus demonstrate a functional association between cellular respiration and vaccine-induced humoral immunity and suggest potential approaches to enhancing vaccine immunogenicity.

**Keywords** Immunometabolism; Mitochondria; OXPHOS; Vaccines; YF17D
**Subject Categories** Immunology; Microbiology, Virology & Host Pathogen Interaction

## Introduction

Vaccination has played an important role in preventing infectious diseases. Vaccination programs have eradicated smallpox and reduced morbidity and mortality from diseases such as poliomyelitis, measles, and, more recently, COVID-19 (Henderson, 2011; Pollard and Bijker, 2021; Zheng et al, 2022). However, despite their overall effectiveness, efforts to develop new and better vaccines remain needed, as underscored by the challenges faced in vaccine development against diseases such as HIV/AIDS (Ng'uni et al, 2020). In part, any vaccine design and development effort faces the obstacle of incomplete understanding of the factors that functionally shape adaptive immune development.

To enable a more granular understanding of the immune response to vaccination, studies in human volunteers have identified molecular correlates of vaccine immunogenicity. In particular, activation and communication between innate and adaptive immune cells, through cytokine and other signaling molecules, are needed to produce immunity (Chan et al, 2016; Hagan et al, 2022; Li et al, 2014; Querec et al, 2009). Additionally, studies with the inactivated trivalent influenza vaccine (TIV) identified plasmablasts and mitochondria signatures as correlates of TIV-induced humoral immunity (Avey et al, 2020; Li et al, 2017; Nakaya et al, 2011; Thakar et al, 2015; Tsang et al, 2014). However, whether these correlates are statistical associations or are functionally involved in shaping the adaptive immune response to vaccination, have largely remained undefined.

Among the various correlates of immunogenicity, the mitochondrion is a promising target for enhancing vaccination responses through currently licensed drugs. Mitochondrial

[1]Programme in Emerging Infectious Diseases, Duke-NUS Medical School, Singapore, Singapore. [2]Department of Infectious Diseases, Singapore General Hospital, Singapore, Singapore. [3]Viral Research and Experimental Medicine Centre, SingHealth Duke-NUS Academic Medical Centre, Singapore, Singapore. [4]SingHealth Investigational Medicine Unit, Singapore General Hospital, Singapore, Singapore. [5]Department of Pharmacy, Singapore General Hospital, Singapore, Singapore. [6]Singapore-MIT Alliance for Research and Technology, Antimicrobial Resistance Interdisciplinary Research Group, Campus for Research Excellence and Technological Enterprise, Singapore, Singapore. [7]Center for Quantitative Medicine, Duke-NUS Medical School, Singapore, Singapore. [8]Center for Child, Adolescent and Maternal Health Research, Tampere University, Tampere, Finland. [9]Saw Swee Hock School of Public Health, National University of Singapore, Singapore, Singapore. [10]Department of Translational Clinical Research, Singapore General Hospital, Singapore, Singapore. ✉E-mail: engeong.ooi@duke-nus.edu.sg; jenny.low@singhealth.com.sg

bioenergetics and metabolism have been intricately linked to immune cell activation, differentiation, and effector activity (Angajala et al, 2018; Breda et al, 2019). Indeed, a recent study has demonstrated that cellular respiration via oxidative phosphorylation (OXPHOS) is critical for the development of humoral immunity and the maturation of long-lived antibody-producing plasma cells (Urbanczyk et al, 2022). These experimental findings collectively suggest that the association between mitochondrial signatures and vaccination-induced antibody response could extend beyond mere statistical associations. Using metformin, an anti-hyperglycemic medication with known inhibitory effects on mitochondrial respiration (Kulkarni et al, 2020; LaMoia and Shulman, 2021), we serendipitously demonstrated a functional role of the mitochondria in live-attenuated yellow fever 17D (YF17D) vaccine immunogenicity.

# Results

## Overview of study

This randomized, double-blind, placebo-controlled clinical trial was originally aimed at determining, as a primary endpoint, if prophylactic metformin treatment could lower endoplasmic reticulum (ER) stress to reduce the rate of systemic symptomatic outcome following YF17D vaccination. A total of 32 healthy volunteers between the ages of 23 and 39 years old, were recruited and randomized, after written informed consent, to receive either placebo or 1000 mg metformin twice daily for 7 days, starting 3 days before and ending 3 days after YF17D vaccination. Blood samples were collected at various timepoints before and after vaccination, as shown in Appendix Fig. S1A. All recruited subjects completed the study. Six subjects were excluded from per-protocol analysis as three in the placebo group developed COVID-19 resulting in missed visits for blood sampling, and three subjects in the metformin group had no measurable metformin in their blood just before YF17D vaccination, possibly from non-compliance (Appendix Fig. S1B,C). Appendix Table S1 summarizes the demographics of the volunteers enrolled in the final analysis ($n = 26$, of which 12 received placebo and 14 received metformin).

The sample size of this study was calculated based on previously observed rates of systemic symptomatic events following YF17D vaccination (Chan et al, 2017; Chan et al, 2019; Chan et al, 2016). Systemic symptoms were coded and classified into organ system classes based on Common Terminology Criteria for Adverse Events (CTCAE): CNS, MSK, respiratory, and gastrointestinal (HSS, 2017). However, the trial vaccine inoculum used was significantly lower than those in previous batches (Chan et al, 2016) (Appendix Fig. S2A). Consequently, only five of the 26 subjects (19.2%) reported any systemic symptoms following YF17D vaccination (Appendix Fig. S2B) compared to previously observed rates that ranged from 35.3 to 63.2% (Camacho et al, 2005; Chan et al, 2017; Chan et al, 2019; Lang et al, 1999; Monath et al, 2002; Pfister et al, 2005; Ripoll et al, 2008). Along with the delays imposed by social distancing and restricted access to the trial center during the COVID-19 pandemic lockdown, the original trial was terminated early for futility in reaching the trial's primary endpoint. Exploratory investigations, however, provided new insights into the molecular mechanisms of how this brief course of metformin therapy impacted the YF17D vaccine immunogenicity outcome.

## Metformin altered the relationship between YF17D viremia and vaccine immunogenicity

Consistent with the lower dose of YF17D inoculum, viremia was detected in only 76.9% of the subjects (Fig. 1A), which was lower than what we had previously encountered (Chan et al, 2016). No difference in viremia levels, either at any of the three timepoints measured post-vaccination (Fig. 1B) or as a composite area under the curve (AUC) derived from all four timepoints (Appendix Fig. S3A), was observed in the metformin compared to placebo-treated subjects. The remaining 23.1% ($n = 6$) of subjects exhibited no detectable viremia. Interestingly, these six individuals exhibited an upregulated expression of innate antiviral and interferon pathways at the pre-vaccination baseline compared to those with detectable viremia (Appendix Fig. S3B), consistent with the well-established role of these pathways in restricting viral replication (Katze et al, 2002).

Vaccine viremia has previously been shown to correlate with YF17D vaccination-induced neutralizing antibody titers (Chan et al, 2016). The lower levels and rate of vaccine viremia in this study thus provided us with a unique opportunity to explore neutralizing antibody development in those subjects with no detectable viremia. Although all six subjects with no detectable viremia developed YF-neutralizing antibodies, the plaque reduction neutralization test titers that reduced 50% of the viral inoculum ($PRNT_{50}$), were significantly greater in the subjects treated with metformin compared to placebo (Fig. 1C). In the placebo-treated group, subjects with no detectable viremia showed significantly lower antibody titers compared to those with detectable viremia. In contrast, antibody titers of those with no detectable viremia in the metformin group were comparable with those who were viremic (Fig. 1C). These findings suggest that metformin treatment altered the known relationship between YF17D viremia and neutralizing antibody response to vaccination.

To examine how this brief course of metformin treatment impacted antibody response, we compared the $PRNT_{50}$ titers against viremia (AUC) of both groups separately. Consistent with earlier findings (Chan et al, 2016), placebo-treated subjects showed a positive and significant correlation between $PRNT_{50}$ titers and viremia (AUC) that followed a sigmoidal trend (Fig. 2A). In contrast, this correlation was not observed in the metformin group (Fig. 2B). The altered relationship was most apparent in a subgroup of subjects with low and no detectable viremia, defined as viremia (AUC) less than the computed IC90 of the placebo curve (Fig. 2A). Indeed, metformin-treated subjects with either low or no detectable viremia ($n = 6$) had significantly higher $PRNT_{50}$ titers than their placebo counterparts ($n = 5$) (Fig. 2C). This finding remained significant even after normalizing $PRNT_{50}$ titers to per unit viremia (AUC) to negate any possibility that subtle viremia differences could confound this analysis (Fig. 2D).

Although viremia has also previously been shown to correlate with T cell responses to YF17D vaccination (Akondy et al, 2015), our data suggest that the effect of metformin treatment was limited to B cells. Ex vivo enzyme-linked immunosorbent spot (ELISpot) assay revealed no significant difference in total IFNγ ELISpot

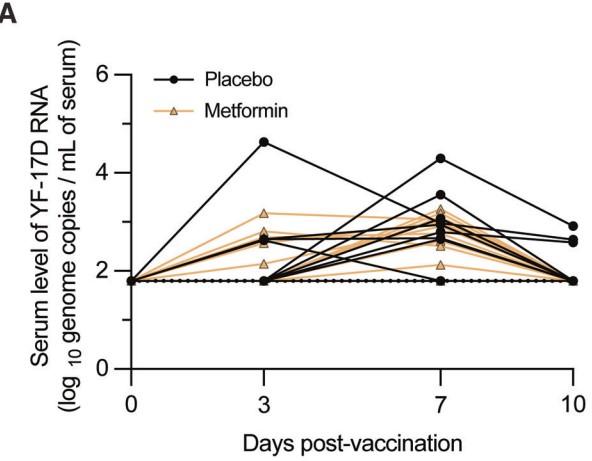

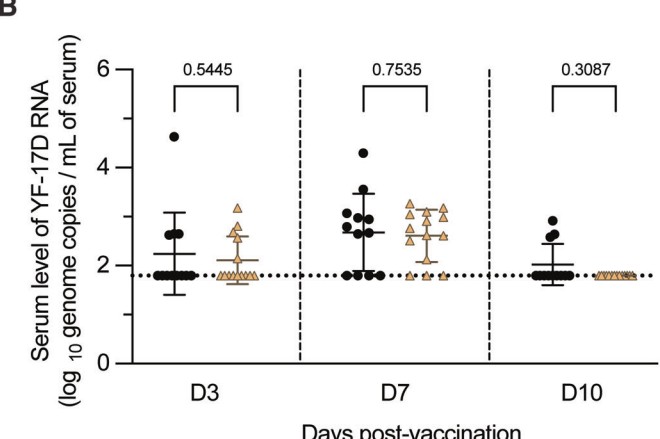

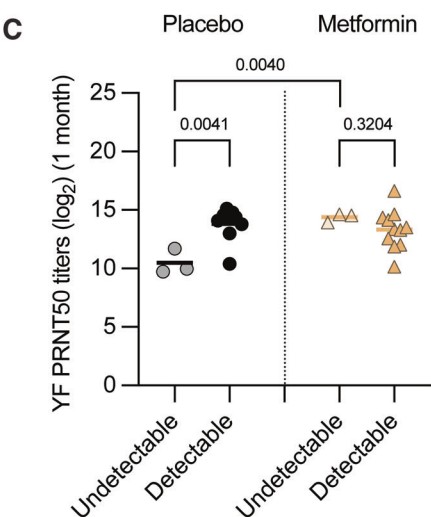

**Figure 1. Metformin alters the correlation between YF17D viremia and neutralizing antibody titers.**

(A) Kinetics of YF17D viremia in placebo and metformin groups at D0 (before vaccination), D3, D7, and D10 post-vaccination as measured by qRT-PCR. (B) Comparison of YF17D viremia at D3, D7, and D10 post-vaccination between placebo ($n = 12$) and metformin ($n = 14$) groups. The dotted line represents the lower limit of detection of the assay. (C) Comparison of YF-neutralizing antibody titers between subjects with no detectable viremia (placebo, $n = 3$; metformin, $n = 3$) versus detectable viremia (placebo, $n = 9$; metformin, $n = 11$) by placebo and metformin groups. Data information: In (B and C), data were presented as mean ± SD. Statistical analyses were performed using an unpaired Student's t-test. Source data are available online for this figure.

counts, when considered using both the whole proteome or by specific viral proteins, between the placebo and metformin subgroups at all the three timepoints measured (Appendix Fig. S4A–C). Neither was there any difference in IFNγ ELISpot counts between subjects with no detectable and detectable viremia (Appendix Fig. S4D).

**Enhanced YF17D neutralizing antibody response was not due to differences in cytokine expression**

To gain insight into the potential mechanisms by which metformin enhances YF-neutralizing antibody response, we first evaluated the plasma concentrations of chemokines and cytokines in the placebo and metformin subgroups (Appendix Table S2). As expected, increased expression of several cytokines, including IFNγ, CXCL10,

CCL2, CCL3, and TNF that peaked at D7 following YF17D vaccination were found in both the placebo and metformin subgroups (Appendix Fig. S5A–E). However, there was no significant difference in cytokine expression at all sampled timepoints between the two subgroups (Appendix Fig. S5F), indicating that the observed enhanced neutralizing antibody response was not driven by changes in cytokine expression.

**Metformin treatment altered the expression of mitochondria and protein translation pathways**

We next used bulk RNA sequencing (RNAseq) to profile the whole blood transcriptome of all subjects with low or no detectable viremia. However, due to issues with sample collection in one subject, RNAseq was performed in five subjects, each from

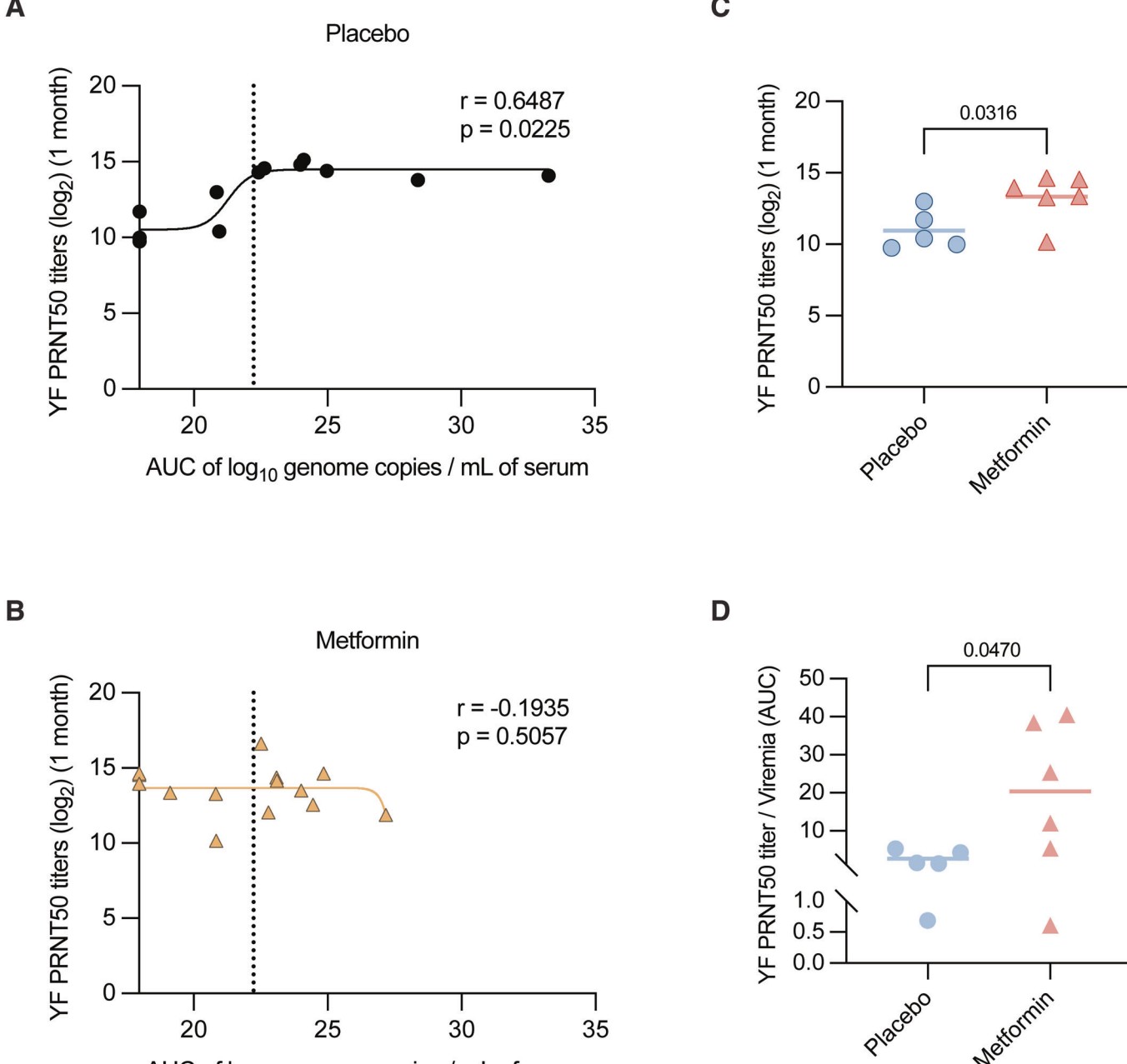

**Figure 2. Metformin augments neutralizing antibody response in YF17D vaccinees with no detectable or low viremia.**

(**A, B**) Relationship between YF-neutralizing antibody titers and viremia (AUC) in placebo-treated and metformin-treated subjects. R values indicate Pearson's correlation between PRNT$_{50}$ titers and viremia (AUC) with the corresponding *p* values. The dotted line in (**A**) represents the IC90 of the placebo curve. This IC90 value (22.23) was used as a cutoff to define a subgroup of subjects with low or no detectable viremia (AUC) in both placebo and metformin groups. (**C**) YF-neutralizing antibody titers of subjects with low or no detectable viremia in the placebo ($n = 5$) and metformin ($n = 6$) subgroups. (**D**) Comparison of PRNT$_{50}$ titers normalized against viremia (AUC) between placebo ($n = 5$) and metformin ($n = 6$) subgroups. Data information: In (**C** and **D**), lines represent the mean. Statistical analyses were performed using an unpaired Student's *t*-test. Source data are available online for this figure.

the metformin and placebo subgroups. Immune deconvolution analyses using the quanTIseq method (Finotello et al, 2019) showed that, apart from a marginally higher frequency of myeloid dendritic cells at D7 in the metformin subgroup, there was no significant difference in the frequencies of monocytes, B cells, CD4 + T cells (non-regulatory), and CD8 + T cells between the two groups

(Appendix Fig. S6A–E). Pre-ranked gene set enrichment analyses (GSEA) (Subramanian et al, 2005) revealed seven categories of biological processes that were differentially enriched in the metformin subgroup compared to placebo. These included processes involved in antiviral and interferon response, DNA replication, as well as inflammation and innate immune responses

(Appendix Fig. S7). However, the changes in the enrichment of these pathways were not in sync with the duration of metformin administration. Only enrichment of pathways related to the mitochondrial electron transport chain (ETC), assembly of respiratory chain complexes, and protein translation in the metformin compared to placebo subgroups occurred over the duration of drug therapy from D0 to D3 post-vaccination (Fig. 3A). As metformin is known to inhibit both mitochondrial respiratory complex I function and protein translation respectively (El-Mir et al, 2000; Larsson et al, 2012; Owen et al, 2000), these findings were consistent with the notion of a compensatory increase in gene expression as part of any negative feedback. Moreover, the expression of leading-edge genes within these pathways at D3 post-vaccination was positively correlated with the eventual YF17D neutralizing antibody titers (Appendix Figs. S8, S9).

That these leading-edge genes are involved in supporting antibody response to vaccination can be gleaned from the transient induction of genes in these pathways at D7 post-vaccination, relative to the pre-treatment baseline (-D3), in the placebo subgroup (Fig. 3B,C). Activated B cells have been shown to switch from glycolysis to OXPHOS as an energy source after 3 days following ex vivo stimulation (Price et al, 2018; Waters et al, 2018). Thus, while the placebo group induced OXPHOS-related genes when required, metformin treatment increased expression before demand. That capacity for OXPHOS on D7 exceeded demand is further suggested by the transient reduction in the abundance of induced genes in the metformin subgroup, before they were upregulated again at D10 (Fig. 3B,C). Finally, the metformin but not placebo subgroup showed increased expression of immunoglobulin genes from D7 to D10 (Fig. 3D; Appendix Table S3).

## Metformin-induced upregulation of ETC genes enabled heightened OXPHOS without oxidative stress

The functional effects of the increased expression of ETC-related genes could be gleaned from the time-dependent changes in plasma polar metabolites in the subgroups. We chose to focus on polar metabolites, given the known function of metformin in controlling glycemia through inhibiting OXPHOS and increasing glycolysis (Foretz et al, 2014; Yang et al, 2021). Significant reduction in the levels of pyruvate, lactate, citrate, and isocitrate were observed in the placebo but not metformin subgroup (Fig. 4A,B) between D3 and D7 post-vaccination, which corresponds to the period during which activated B cells switch from glycolysis to OXPHOS (Price et al, 2018; Waters et al, 2018). By contrast, other glycolytic and tricarboxylic acid (TCA) cycle metabolites were not significantly different between D3 and D7 post-vaccination in both subgroups (Fig. 4B; Appendix Fig. S10).

Pyruvate and lactate were recently reported to promote resistance to reactive oxygen species (ROS)-mediated oxidative stress (Tauffenberger et al, 2019). Hence, the lower levels of pyruvate and lactate at D7 compared to D3 in the placebo subgroup suggest the possibility of increased ROS production during the period where OXPHOS activity is increased (Liu et al, 2002); inefficient transfer of electrons from one respiratory complex to another allows free electrons to react with oxygen to form superoxide radicals (Raimondi et al, 2020). The absence of any reduction in pyruvate and its downstream metabolites in the metformin subgroup thus suggests that the increased capacity for

OXPHOS enabled more efficient electron transfer. This notion is further supported by the changes in expression, from D3 to D7 post-vaccination, of ROS scavenger enzymes. Increased expression of cytosolic superoxide dismutase 1 (SOD1) expression was found in the placebo but not metformin subgroup (Fig. 4C). On the other hand, mitochondrial SOD2 expression was reduced in the metformin but not placebo subgroup (Fig. 4C). Elevated expression of another key antioxidant defense enzyme glutathione reductase (GR) (Couto et al, 2016), was observed in the placebo subgroup but not in the metformin subgroup. Glutathione synthetase (GSS) expression was, however, unchanged in both subgroups (Fig. 4D). Taken collectively, our data suggest that metformin treatment expanded the abundance of ETC-related proteins that, upon treatment cessation at D3, allowed for OXPHOS to occur in activated B cells with minimal mitochondrial oxidative stress.

## Effect of short-course metformin on YF17D vaccination can be reproduced in C57BL/6 mice

If indeed metformin expanded OXPHOS capacity that, in turn, enhanced antibody response to YF17D, these findings should be reproducible in an animal model. We chose C57BL/6 mice to test this possibility as this mouse strain does not support viremic YF17D infection (Erickson and Pfeiffer, 2015), thus representing the human subjects where enhanced antibody response to vaccination were observed. We also compared the impact of a 7-day course against an extended course of metformin on neutralizing antibody responses. If OXPHOS activity after D3 post-vaccination led to higher YF17D neutralizing antibody titers, the extended course of metformin and hence OXPHOS inhibition would reduce antibody titers.

C57BL/6 mice were administered 4 mg of metformin twice daily (equivalent to our human experimental dose of 1000 mg twice daily) (Nair and Jacob, 2016), starting from 3 days before vaccination until either D3 (7-day course) or D10 (extended course) after vaccination, and serum was collected at D21 post-vaccination for antibody measurement (Fig. 5A). With an inoculum of $10^4$ PFU of YF17D, four out of five (80%) mice that received a 7-day course of metformin seroconverted, with seroconversion rates that were significantly higher than both untreated animals as well as those on an extended course of metformin (Fig. 5B). Similarly, YF17D $PRNT_{50}$ titers were highest in mice given a 7-day course of metformin than the other two groups, although the comparison with the extended treatment group did not reach statistical significance (Fig. 5C). Neutralizing antibody titers produced from a higher YF17D dose ($10^5$ PFU) were comparable in all three groups (Fig. 5D), again supporting our clinical findings where the benefit of short-course metformin was limited to only the subgroup with low or undetectable viremia.

## Discussion

How cellular processes drive adaptive immune response to vaccination, and even infection, remains incompletely defined despite the consistent association between these processes and the magnitude of adaptive immunity. In this randomized, double-blind, placebo-controlled healthy volunteer study, we serendipitously revealed how modulation of mitochondrial activity with metformin

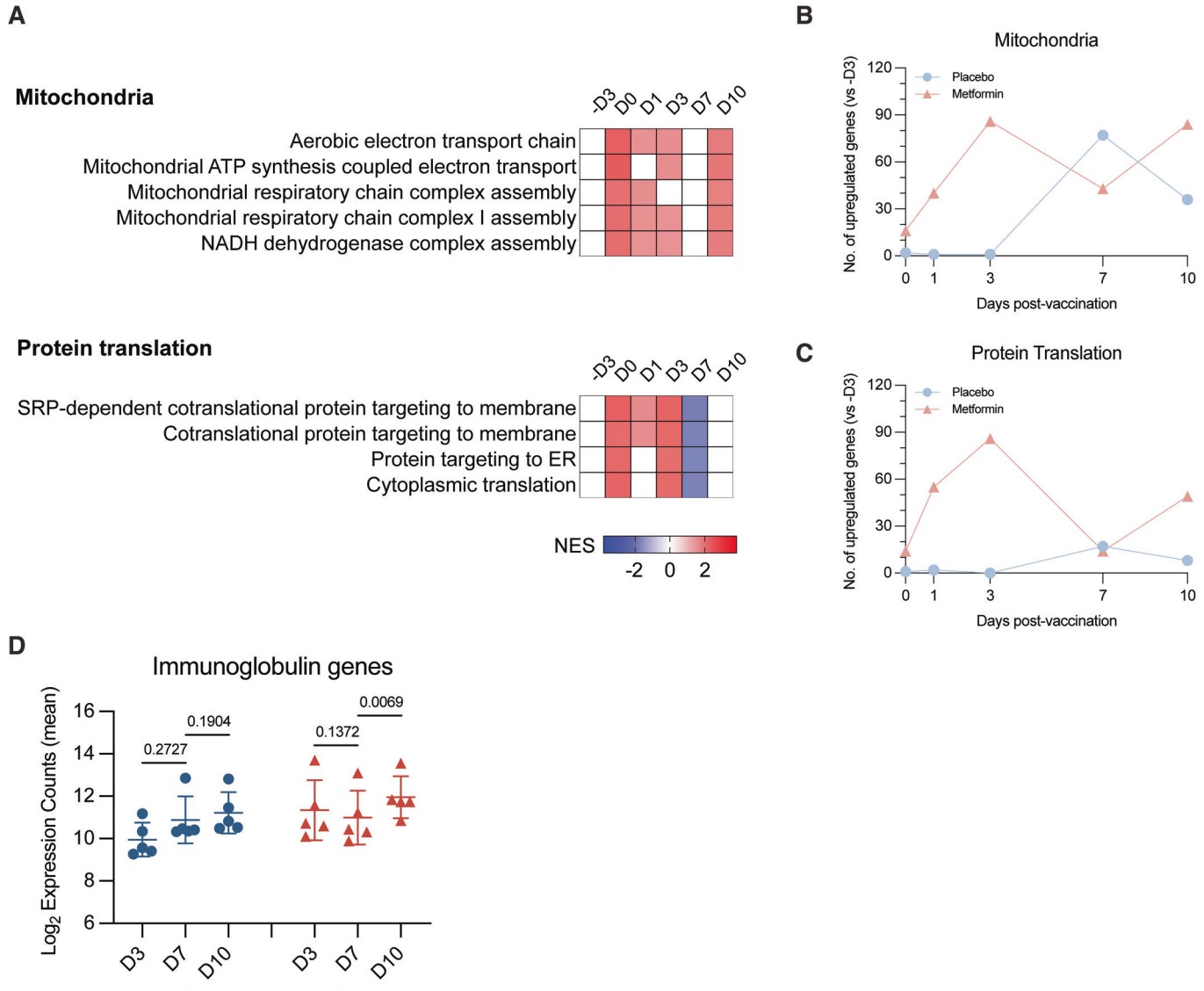

**Figure 3. Increased enrichment of mitochondria- and protein translation-related pathways in metformin compared to the placebo subgroup.**

(A) Gene set enrichment analysis (GSEA) pre-ranking with Gene Ontology Biological Process (GOBP) module at baseline (-D3), before vaccination (D0), D1, D3, D7, and D10 post-vaccination as represented by normalized enrichment scores (NES). Red, blue, and white colors indicate increased, decreased, or no significant difference in enrichment, respectively, between metformin compared to placebo subgroups. (B, C) Number of upregulated mitochondrial leading-edge genes and protein translation leading-edge genes from significantly enriched pathways identified in (A) at D0, D1, and D3 post-vaccination. A gene was considered upregulated if its average fold change (vs baseline) of all subjects in the subgroup was >1. (D) Mean log$_2$ expression counts of immunoglobulin genes in placebo (blue circles) and metformin (red triangles) subgroups at D3, D7, and D10 post-vaccination. Each data point depicts a subject-derived sample. The list of genes analyzed is summarized in Appendix Table S3. Data were represented as mean ± SD. Statistical analyses were performed using paired two-tailed *t*-test. Source data are available online for this figure.

can enhance vaccine immunogenicity. We found that a brief course of metformin treatment, given 3 days before, to 3 days after YF17D vaccination, resulted in significantly higher neutralizing antibody titers compared to placebo recipients, albeit limited to those who developed low or no detectable vaccine viremia. The enhanced antibody response was independent of changes in cytokine expression, and innate immune and lymphocyte counts. Instead, it correlated with the expression of genes involved in protein translation, mitochondrial respiratory complexes, as well as the assembly factors for these complexes.

The increased expression of respiratory complex transcripts from metformin treatment is likely a negative feedback response to the metformin-induced inhibition of respiratory complex I. Indeed, a recent cross-sectional study of Type 2 Diabetes Mellitus (T2DM) patients found increased protein expression of mitochondrial respiratory complexes in PBMCs of metformin users compared to non-users (de Marañón et al, 2022). Stopping of metformin at D3 post-vaccination coincided with the known timepoint where B cells upregulate OXPHOS following ex vivo stimulation (Price et al, 2018). Release of inhibition by metformin on the accumulated

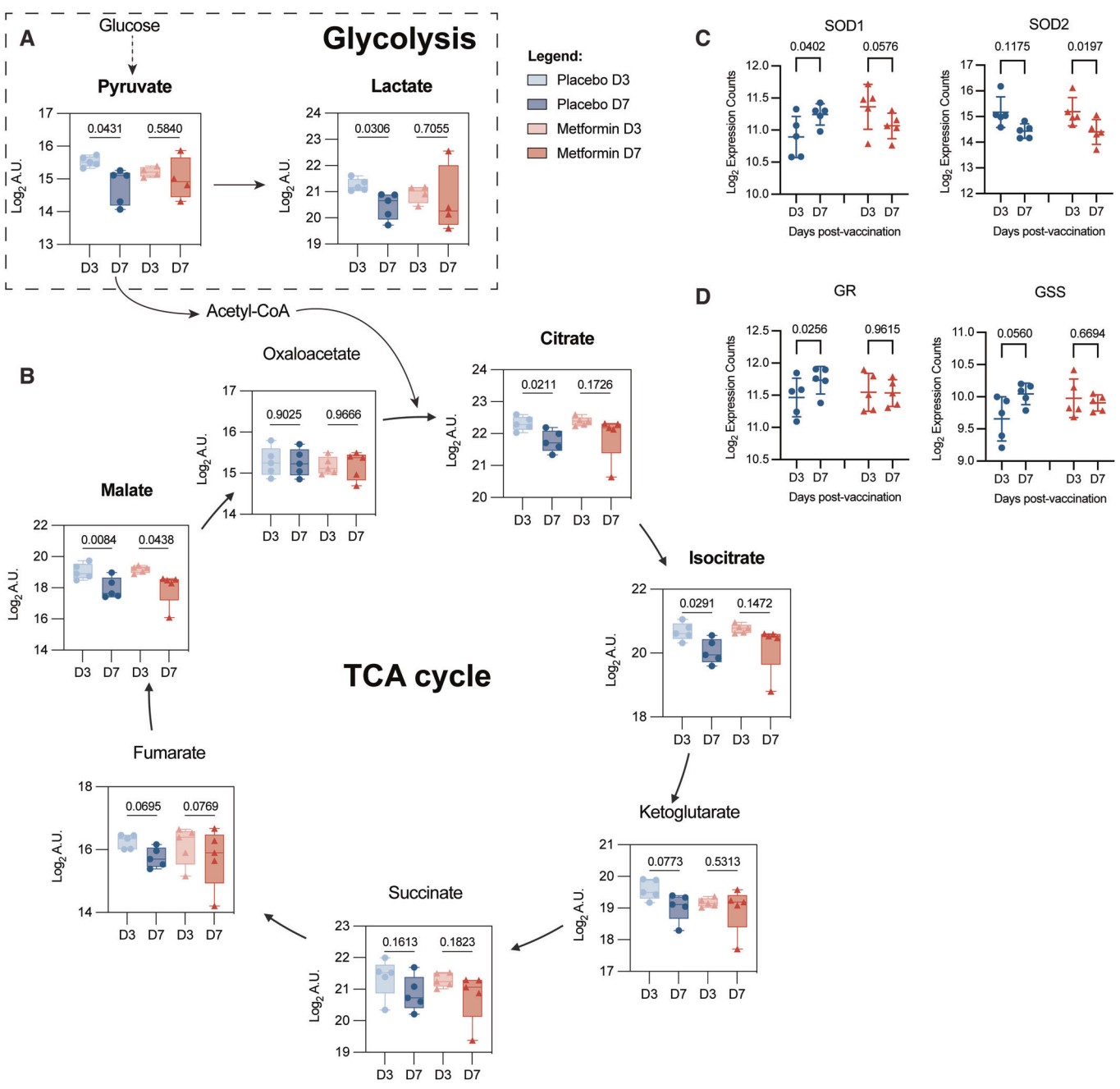

**Figure 4. Alterations in metabolites suggest short-term metformin treatment reduced oxidative stress.**

(A) Plasma levels of pyruvate and lactate in placebo ($n = 5$) and metformin ($n = 4$) subgroups at D3 and D7 post-vaccination. (B) Plasma levels of TCA cycle intermediates in placebo ($n = 5$) and metformin ($n = 5$) subgroups at D3 and D7 post-vaccination. (C, D) RNAseq gene expression of cytosolic superoxide dismutase (SOD1), mitochondrial superoxide dismutase (SOD2), and glutathione reductase (GR) and glutathione synthetase (GSS) at D3 and D7 post-vaccination. Blue circles and red triangles represent placebo-treated and metformin-treated subjects, respectively. Data information: In (A and B), data were represented as box plots showing all data points. The lines represent medians, boxes represent 25th to 75th percentile intervals, and whiskers represent 5th to 95th percentile intervals. In (C and D), data were represented as mean ± SD. Plasma metabolite levels are represented as $log_2$ transformed arbitrary units (A.U.). Statistical analyses were performed using paired two-tailed $t$-test. Source data are available online for this figure.

expression of respiratory complex transcripts in B cells at this point could have thus supported more efficient OXPHOS and hence antibody expression. Indeed, the importance of OXPHOS in the development of humoral immunity has previously been shown: Treatment with dichloroacetate, a pyruvate kinase inhibitor, to

promote OXPHOS at the expense of glycolysis, resulted in increased differentiation of naïve B cells into plasmablasts following ex vivo stimulation (Price et al, 2018); inhibition of ATP synthase with oligomycin, in contrast, prevented differentiation of naïve B cells into plasmablasts and immunoglobulin class switching to IgG1

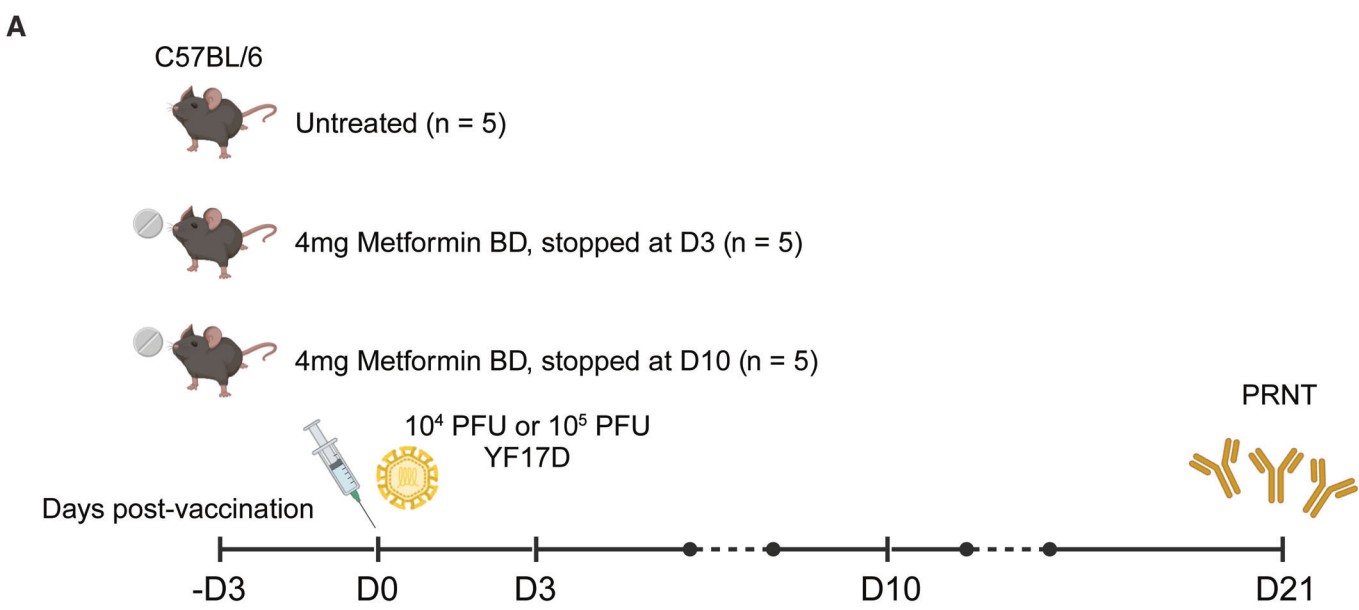

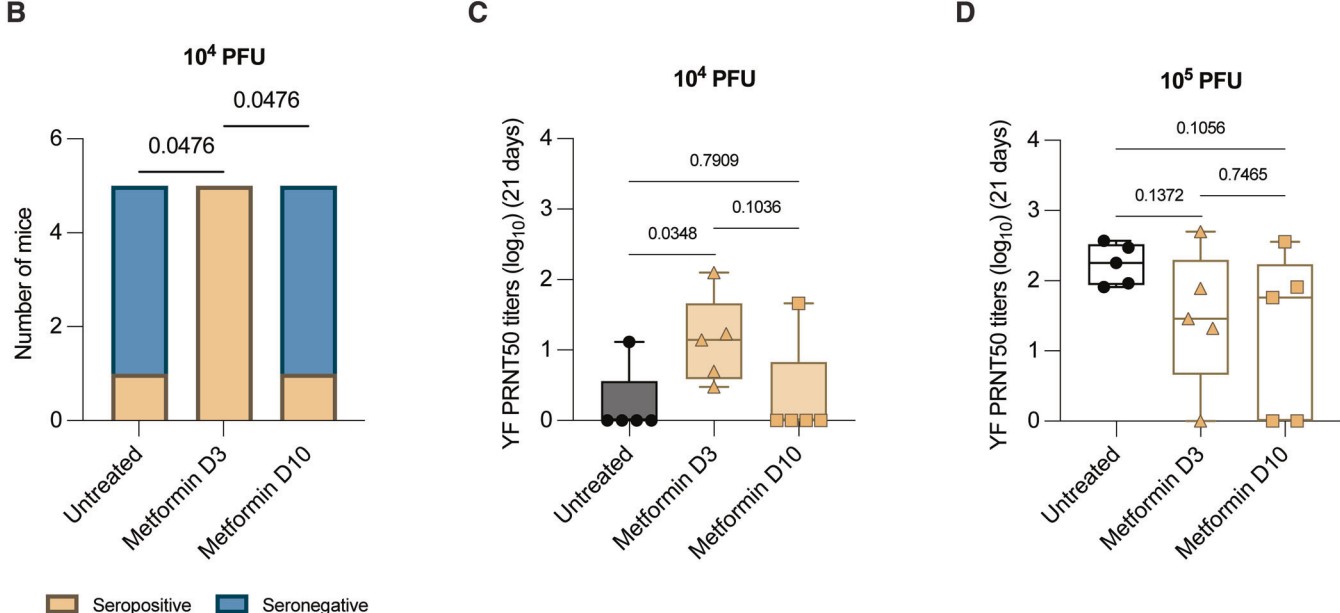

**Figure 5. Cessation of metformin therapy at D3, but not D10 post-vaccination, enhances YF17D vaccination outcomes in C57BL/6 mice.**

(A) Schematic illustration of the experiment design. 6 to 8-week-old C57BL/6 mice were treated with water (untreated) or 4 mg of metformin twice daily by oral gavage. All mice were vaccinated with either $10^4$ plaque-forming units (PFU) or $10^5$ PFU of YF17D after 3 days of treatment (D0), and metformin treatment was continued until D3 or D10 post-vaccination. Schematic was created with BioRender.com. (B) Fisher's exact test comparing the proportion of mice that seroconverted following vaccination with $10^4$ PFU of YF17D. (C, D) YF-neutralizing antibody titers in mice vaccinated with $10^4$ or $10^5$ PFU of YF17D at D21 post-vaccination. Data information: In (C and D), data are represented as box plots showing all data points. The lines represent medians, boxes represent 25th to 75th percentile intervals, and whiskers represent 5th to 95th percentile intervals. Statistical analyses were performed with unpaired Student's *t*-test. Source data are available online for this figure.

(Waters et al, 2018); suppression of OXPHOS in ex vivo-generated plasmablasts with rotenone (complex I inhibitor), antimycin A (complex III inhibitor) or oligomycin resulted in decreased antibody secretion (Price et al, 2018). A brief course of metformin

before and after YF17D vaccination on B cells could thus be likened to "expanding a highway before the surge in traffic".

Numerous factors, including host genetics and environmental influences, have been shown to influence vaccination responses

(Zimmermann and Curtis, 2019). More recently, studies have demonstrated that alterations in gut microbiota composition and microbiota-produced short-chain fatty acids (SCFAs) levels can impact vaccination-induced humoral responses (Hagan et al, 2019; Kim et al, 2016). Interestingly, metformin therapy has been linked to changes in the gut microbiome and SCFAs levels (Mueller et al, 2021; Rosell-Díaz and Fernández-Real, 2023). However, our animal data suggest a more direct role for metformin in shaping vaccination-induced humoral responses. Notably, cessation of metformin at the critical timepoint of D3, but not D10 post-vaccination, resulted in enhanced antibody responses following live-attenuated YF17D vaccination.

The mechanism behind how metformin selectively enhances humoral, but not cellular immune responses after live-attenuated YF17D vaccination is unclear at this point. One possible explanation for these findings may lie in the distinct metabolic requirements of B and T cell activation. B cells upregulate both glycolysis and OXPHOS upon activation, while T cells preferentially undergo a metabolic switch from OXPHOS to glycolysis (Sack, 2018; Waters et al, 2018). Furthermore, both glucose deprivation and inhibition of glycolysis have been shown to impair IFNγ production in both activated CD4+ and CD8+ T cells, in vitro (Cham et al, 2008; Chang et al, 2013). Additional investigations are required to elucidate the precise impact of metformin on the kinetics of both humoral and cellular immune responses, particularly during the early stages following vaccination. Single-cell assays, including scRNAseq or Met-Flow (Ahl et al, 2020), may help to shed deeper insights into the complex interplay between metabolism and adaptive immunity.

Finally, our findings have important clinical implications. Globally, metformin is the most commonly prescribed oral anti-hyperglycemic agent to manage T2DM (Clemens et al, 2020). It is also being actively explored for other indications, including as an anti-aging agent (Kulkarni et al, 2020). Long-term metformin may, however, compromise humoral immune responses to vaccination (Sun et al, 2020). Indeed, our findings in YF17D vaccinated C57BL/6 mice call for this possibility to be examined expediently, especially since T2DM patients and the elderly are both at increased risk of severe infections (Casqueiro et al, 2012; Mueller et al, 2020). Although our study used YF17D exclusively, our findings may extend to other forms of vaccine: Clinical studies in T2DM patients reveal that patients on chronic metformin therapy exhibit diminished antibody responses following TIV vaccination compared to non-metformin users, irrespective of their glycemic control (Agarwal et al, 2018; Saenwongsa et al, 2020). Collectively, these findings suggest that stopping metformin after vaccination or substituting metformin with other anti-hyperglycemic agents temporarily may improve antibody response to vaccination in those on long-term metformin treatment.

In conclusion, mitochondrial respiration is functionally associated with antibody response to live-attenuated YF17D vaccination.

# Methods

## Clinical trial design

Thirty-seven healthy adults, who have not previously received YF17D vaccination, were pre-screened for anti-dengue antibodies (Panbio DENV IgG, Abbott) and enrolled upon written informed consent. Participants were then randomized 1:1 to receive either placebo (vitamin D/calcium tablet) or 1000 mg metformin twice daily from 3 days before to 3 days after live-attenuated YF17D vaccine administration (Stamaril, Sanofi-Pasteur) [Lot # R3P101V, Expiry: 11/2021; Lot # T3F381V, Expiry: 05/2022; Lot # U3F416V, Expiry: 04/2023]. Serum and plasma were collected on -D3 (before placebo/metformin dosing), D0 (pre-vaccination), D3, D7, D10, and 1 month after vaccination for analysis. PBMCs were collected at -D3, D0, D14, and 1 month post-vaccination. Whole blood was extracted at -D3, D0, D1, D3, D7, and D10 post-vaccination and stored in Tempus Blood RNA tubes (Thermo Fisher, catalog number: 4342792) at −80 °C until use. Throughout the study period, all local and systemic symptoms experienced up to one month following placebo/metformin treatment were collected.

## Measurement of metformin levels

Plasma concentrations of metformin were measured using an in-house developed method on a Shimadzu Nexera X2 ultra-high-performance liquid chromatography system (uHPLC) interfaced with a Shimadzu photodiode array module (Shimadzu, Kyoto, Japan). Chromatographic separation and detection were achieved using an Acquity UPLC BEH amide column, 1.8 μm, (2.1 × 100 mm) (Waters, Milford, MA, USA). Chromatographic peak integration was performed using the Lab Solutions Software (v5.85, Shimadzu). Study plasma and/or metformin-spiked plasma calibrator samples were extracted via protein precipitation with acetonitrile and centrifuged. The resulting supernatant was mixed with a mobile phase and passed through a filter prior to injection into the system. A total of seven calibration levels and three controls (low: LQC, mid: MQC, and high: HQC) were prepared using analytic grade metformin (Toronto Research Chemicals) to construct the calibration curve. Accuracy and precision were considered acceptable if they were within ±15% in accordance with the 2018 Bioanalytical Method Validation Guidance for Industry by the US Food and Drug Administration (FDA) (FDA, 2018).

## Cells and viruses

Vero and BHK-21 were purchased from ATCC, and all cell lines were tested for mycoplasma contamination. YF17D virus was isolated from Stamaril by passaging once in Vero cells. Infectious titer was determined by plaque assay (Chan et al, 2016), and viruses were stored at −80 °C until use. All experiments with the YF17D virus were performed in a biosafety level 2 laboratory with approval from the Office of Safety, Health and Environment (OSHE), National University of Singapore (NUS).

## YF17D viremia assessment with qRT-PCR

Viral titer was determined by qRT-PCR and a standard curve was constructed to quantitate viral copy number as previously described (Low et al, 2020). In brief, viral RNA was extracted from sera using the Roche MagNA Pure 24 Total NA isolation kit, followed by real-time quantitative reverse transcription polymerase chain reaction (Invitrogen SuperScript® III One-Step Quantitative RT-PCR) according to the manufacturer's protocol. Raw quantitation cycle (Cq) values were recorded, and a cycle threshold value of 40 was

used as the limit of detection. Raw Cq values were subsequently converted into $\log_{10}$ genome copies per mL for reporting purposes. The standards used for conversion were derived from YF17D RNA standards that were quantified with a known copy number per mL. Technical duplicates were performed for each standard concentration, and the values of eight independent runs were consolidated to construct the standard curve for conversion. Viremia (AUC) was calculated with Prism 9.4.0 software (GraphPad Software Inc.) using 0 as a baseline. The sequence of the YF17D-specific primers used are as follows:

YF17D Forward: 5′ GAACAGTGATCAGGAACCCTCTCT 3′;
YF17D Reverse: 5′ GGATGTTTGGTTCACAGTAAATGTG 3′;
YF17D Probe: 5′ HEX-CTACGTGIC/ZEN/TGGAGCCCG-CAGCAAT-IBFQ 3′.

## Plaque reduction neutralization test

Plaque reduction neutralization test (PRNT) was performed on BHK-21 cells (ATCC) as previously described using sera obtained on day 0 and 1 month after YF17D administration (Chan et al, 2016). In brief, serial two-fold dilutions of sera in RPMI maintenance media (RPMI-MM) were incubated with 40 plaque-forming units (PFU) of YF17D virus for one hour at 37 °C followed by addition to BHK-21 cells (100 μL per well). The mixture was aspirated after incubation for 1 h at 37 °C. Cells were then overlaid with 1% carboxymethylcellulose (CMC) in RPMI-MM and incubated for 5 days at 37 °C. Subsequently, cells were fixed with 20% formaldehyde and stained with 1% crystal violet. $PRNT_{50}$ titers were calculated using a sigmoid dose-response curve and reported as reciprocal values.

## Transcriptomic profiling with RNAseq

Whole blood was collected from volunteers and stored in Tempus blood RNA tubes (Thermo Fisher) at −80 °C as described above. Samples were thawed at room temperature, and RNA isolation was carried out with the Tempus spin RNA isolation kit, eluted in 90 μL of RNase-free water in accordance with the manufacturer's protocol (Thermo Fisher, catalog number: 4380204). RNA sequencing was performed with Novaseq PE150 lane sequencing at Azenta Life Sciences and deposited onto ArrayExpress.

## RNAseq analysis and gene set enrichment analysis (GSEA)

Quality control of RNAseq data was performed using the decoupleR method with a threshold of 4 (Badia-i-Mompel et al, 2022), and unannotated reads were removed prior to analysis. Partek software was used to calculate the difference in gene expression at each sampled timepoint (-D3, D0, D1, D3, D7, and D10 post-vaccination) by computing the difference between RNAseq $\log_2$ gene expression counts of the metformin subgroup and the placebo subgroup (metformin - placebo). This difference was then subjected to GSEA pre-ranking (GSEA Java software version 4.3.0, Broad Institute) against the Gene Ontology Biological Process (GOBP) 2021 and the Blood Transcription Module (BTM) (2013 version) database. Gene sets were considered significantly enriched if their FDR q-value is less than 0.05. To create the

heatmaps of pathway enrichment, we initially identified the top ten most positively and negatively enriched pathways at each sampled timepoint based on both the magnitude of the normalized enrichment score (NES) and an FDR q-value less than 0.05. Subsequently, we then determined if the identified pathways were significantly enriched across all timepoints. If a pathway had an FDR q-value less than 0.05, we plotted the NES for that specific timepoint. Gene expression heatmaps were plotted with the z-score normalized expression of leading-edge genes in differentially enriched pathways. Leading-edge genes were identified based on the definition described by Subramanian et al (2005), which described these genes as "the genes of a gene set that account for the enrichment signal". Heatmaps were created using Prism software (GraphPad Prism Inc.).

## Olink cytokine profiling

Plasma cytokine levels were measured using Olink Target 48 Cytokine panel as described previously (Ong et al, 2023). Briefly, serum samples were first incubated with proximity antibody pairs tagged with unique DNA reporter oligonucleotides. Upon binding to target proteins, the antibody pairs come into proximity, leading to hybridization and extension of the DNA reporter barcode. The double-stranded DNA barcode was then amplified and quantified using the Olink Signature Q100. Cytokines with missing data frequency greater than 75% were excluded from the final analysis.

## PBMC isolation

Peripheral blood was collected in EDTA-containing tubes and PBMCs were isolated using SepMate-50 tubes according to the manufacturer's protocol (STEMCELL Technologies). Isolated PBMCs were stored at −80 °C for at least 24 h, after which they were stored in liquid nitrogen until use.

## Quantification of YF17D-specific T cells

To quantify the frequency of YF17D-specific T cells, cryopreserved PBMCs were stimulated with pools of 15-mer peptides overlapping by ten amino acids that span the structural (capsid, pre-membrane, envelope) and non-structural (NS1 to NS5) proteins of YF17D in an IFNγ ELISpot assay. Briefly, ELISpot plates (Merck) were coated overnight at 4 °C with human IFNγ antibody (Mabtech, 3420-2-1000). A total of 200,000 PBMCs were seeded per well and stimulated overnight with the YF17D peptide pools at 1 μg/mL. The plates were subsequently incubated with a biotinylated anti-human IFNγ antibody (Mabtech, 3420-6-1000), followed by streptavidin-alkaline phosphatase (Mabtech, 3310-10-1000) and developed using a BCIP/NBT phosphatase substrate (Seracare). To quantify positive peptide-specific responses, mean spots of the unstimulated wells were subtracted from the peptide-stimulated wells, and the results were expressed as spot-forming units (SFU) per $10^6$ PBMCs.

## Measurement of plasma metabolites

Plasma levels of glycolytic and TCA cycle metabolites were measured and analyzed by a liquid chromatography–tandem mass spectrometry (LC-MS/MS) protocol as previously described with modifications

(Zhong et al, 2017). In brief, a volume of 50 µL from each plasma sample was thawed at 4 °C, and plasma proteins were precipitated with 200 mL of ice-cold methanol. After vortexing, the mixture was centrifuged at 16,000 rpm for 10 min at 4 °C, and the supernatant was collected and evaporated to dryness in a vacuum evaporator. The dry extracts were then redissolved in 100 µL of 0.1% formic acid in water for LC-MS/MS analysis. Quality control (QC) samples were prepared by mixing equal amounts of serum samples from all the samples and processed as per other samples. The QC sample was run after each eight samples to monitor the stability of the system, and all samples were randomized. LC-MS/MS analysis was conducted using an Agilent 1290 ultrahigh pressure liquid chromatography system coupled to a 6495 Triple Quadrupole mass spectrometer equipped with a dual-spray electrospray ionization source with Jet Stream™. Chromatographic separation of glycolysis intermediates and organic acid was achieved by using Phenomenex Rezex™ ROA-Organic Acid H+ (8%) column (2.1 × 100 mm, 3 µm) and the compounds were eluted at 40 °C with an isocratic flow rate of 0.3 mL/min of 0.1% formic acid in water. All the metabolites were quantified in multiple reaction monitoring (MRM) mode, and mass transition and collision energy were optimized for each compound by direct infusion of individual standard solutions. Electrospray ionization was performed in negative ion mode with the following source parameters: drying gas temperature 300 °C with a flow of 10 L/min, nebulizer gas pressure 40 psi, sheath gas temperature 350 °C with a flow of 11 L/min, nozzle voltage 500 V, and capillary voltage 3000 V. Data acquisition and processing were performed using Agilent MassHunter software.

## Animal studies

C57BL/6 mice were purchased from InVivos Pte Ltd and housed in Duke-NUS Medical School. Female 6- to 8-week-old mice were used for vaccination studies. All animals were maintained in accordance with protocols approved by the Institutional Animal Care and Use Committee at Singapore Health Services, Singapore (ref no.: 2021/SHS/1658).

Mice were treated twice daily with 100 µL of sterile water (untreated) or 4 mg of metformin dissolved in 100 µL of sterile water by oral gavage. This animal equivalent dose (AED) of metformin (4 mg) was calculated in accordance with previously described methods (Nair and Jacob, 2016) based on the human dosage (1000 mg) used in the current clinical study. After 3 days of treatment, all mice were intraperitoneally (i.p.) vaccinated with either $10^4$ PFU or $10^5$ PFU of YF17D virus diluted in 100 µL of sterile PBS. Metformin treatment was continued after vaccination and was stopped at either D3 or D10 post-vaccination. At D21 post-vaccination, all mice were sacrificed to acquire serum for PRNT analyses as described above.

## Statistics

The clinical trial sample size was calculated using a randomized screening trial design for prioritization of new treatments as candidates for future phase III evaluation (Rubinstein et al, 2005). Subject randomization was performed using an online randomization tool available at http://www.randomization.com by an independent individual who has no direct contact with the subjects. Opaque envelopes, prepared by an independent team according to the Master

Randomization List, were opened sequentially by the study team to assign subjects to the treatment arms. For in vitro experiments, no blinding or randomization occurred, and all samples were run in duplicates. For animal studies, all animals were included in the final analysis (no exclusion). All statistical analyses were performed using Prism 9.4.0 software (GraphPad Software Inc.). Statistical significance was determined using an unpaired, two-tailed Student's $t$-test, after ensuring data were normally distributed using Kolmogorov–Smirnov test at alpha = 0.05. For paired data, the paired Student's $t$-test was performed. Data were represented as mean ± S.D.

## Study approval

Approval for this clinical study was obtained from the SingHealth Centralized Institutional Review Board (ID:2021/2509) and is registered under clinicaltrials.gov registration number NCT04267809. The experiments involving human participants conformed to the principles set out in the WMA Declaration of Helsinki and the Department of Health and Human Services Belmont Report. All participants gave written informed consent prior to enrollment.

## For more information

- FOXRED1 OMIM entry: https://www.omim.org/entry/613622?search=FOXRED1&highlight=foxred1
- NDUFA13 OMIM entry: https://www.omim.org/entry/609435?search=ndufa13&highlight=ndufa13

---

### The paper explained

**Problem**

While vaccination has effectively controlled many infectious diseases, there remains an ongoing need to develop better and more effective vaccines. Research examining the host response to vaccination have identified several host factors including mitochondrial related factors, that are correlated with vaccine immunogenicity. Distinguishing statistical correlates from functional associations is thus needed as only functional associations would provide mechanistic insights into vaccine-induced adaptive immunity.

**Results**

In a randomized-controlled healthy volunteer study, we found that a brief course of metformin, started shortly before yellow fever 17D (YF17D) vaccination followed by timely cessation 3 days post-vaccination, led to higher YF17D neutralizing antibody titers compared to placebo. Gene expression and metabolite analyses demonstrated that this short course of metformin bolstered oxidative phosphorylation and protein translation capacities, which correlated with higher neutralizing antibody levels without excessive triggering of the reactive oxygen species response. The importance of metformin cessation at Day 3 post-vaccination was further validated in a mouse experiment.

**Impact**

Our study suggests that mitochondrial respiration is functionally associated with antibody response to live-attenuated YF17D vaccination. This discovery highlights a potential clinical strategy to enhance antibody response to vaccination in patients on long-term metformin treatment.

- NDUFB6 OMIM entry: https://www.omim.org/entry/603322?search=ndufb6&highlight=ndufb6.

## Data availability

The datasets produced in this study are available in the following databases: Raw RNAseq data and processed log$_2$ expression counts: ArrayExpress, European Molecular Biology Laboratory-European Bioinformatics Institute (EMBL-EBI) (https://www.ebi.ac.uk/arrayexpress), under accession no. E-MTAB-3126 (https://www.ebi.ac.uk/biostudies/arrayexpress/studies/E-MTAB-13265?key=4a75498a-e061-4685-915d-111cd49f3339).

## Peer review information

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

## Acknowledgements

We thank all the volunteers, research coordinators, and nurses at the SingHealth Investigational Medicine Unit for their time and assistance in the conduct of this trial. This study was supported by the Clinician-Scientist Award from the National Medical Research Council of Singapore (NMRC/CSAINV19May-002) awarded to Jenny G. Low. Eng Eong Ooi received salary support from the Singapore Translational Research Award from the National Medical Research Council of Singapore (MOH-001271-00).

## Author contributions

**Darren ZL Mok**: Data curation; Formal analysis; Investigation; Visualization; Methodology; Writing—original draft; Project administration; Writing—review and editing. **Danny JH Tng**: Resources; Investigation. **Jia Xin Yee**: Investigation; Project administration. **Valerie SY Chew**: Investigation. **Christine YL Tham**: Investigation; Methodology. **Justin SG Ooi**: Investigation. **Hwee Cheng Tan**: Investigation. **Summer L Zhang**: Investigation. **Lowell Z Lin**: Investigation. **Wy Ching Ng**: Investigation. **Lavanya Lakshmi Jeeva**: Resources. **Ramya Murugayee**: Resources. **Kelvin K-K Goh**: Investigation; Methodology. **Tze-Peng Lim**: Investigation; Methodology. **Liang Cui**: Investigation; Methodology. **Yin Bun Cheung**: Conceptualization; Methodology. **Eugenia Z Ong**: Resources; Methodology. **Kuan Rong Chan**: Formal analysis. **Eng Eong Ooi**: Conceptualization; Formal analysis; Supervision; Funding acquisition; Methodology; Writing—original draft; Writing—review and editing. **Jenny G Low**: Conceptualization; Resources; Formal analysis; Supervision; Funding acquisition; Methodology; Writing—original draft; Writing—review and editing.

## Disclosure and competing interests statement

The authors declare no competing interests.

