## [Peer Review File · EMBO Molecular Medicine]

Electron transport chain capacity expands yellow fever vaccine immunogenicity

Darren Mok, Danny Tng, Jia Xin Yee, Valerie Chew, Christine Tham, Justin Ooi, Hwee Cheng Tan, Summer Zhang, Lowell Lin, Wy Ching Ng, Lavanya Jeeva, Ramya Murugayee, Kelvin Goh, Tze-Peng Lim, Liang Cui, Yin Bun Cheung, Eugenia Ong, Kuan Rong Chan, Eng Eong Ooi, and Jenny Low

Corresponding authors: Eng Eong Ooi (engeong.ooi@duke-nus.edu.sg), Jenny Low (jenny.low@singhealth.com.sg)

Review Timeline:

Submission Date:	29th Sep 23
Editorial Decision:	15th Nov 23
Revision Received:	22nd Feb 24
Editorial Decision:	22nd Mar 24
Revision Received:	1st Apr 24
Accepted:	2nd Apr 24

Editor: Zeljko Durdevic

Transaction Report:

15th Nov 2023

Dear Dr. Low,

Thank you for the submission of your manuscript to EMBO Molecular Medicine, and please accept my apologies for the delay in getting back to you, which is due to the fact that one referee needed more time to complete his/her review. We have now received feedback from the three reviewers who agreed to evaluate your manuscript. All three referees recognize potential interest of the study but also raise important criticism that should be addressed in a major revision. Confirmation of main findings from human trial (particularly association between viremia, metformin, and antibody neutralization) in a small animal model as suggested by the referee #2 would substantially strengthen the findings and is essential for further consideration of the manuscript in our journal. If you would like to discuss further the points raised by the referees, I am available to do so via email or video. Let me know if you are interested in this option.

We would welcome the submission of a revised version within six months for further consideration. Please let us know if you require longer to complete the revision.

I look forward to receiving your revised manuscript.

Yours sincerely,

Zeljko Durdevic

We require:

- 1) A .docx formatted version of the manuscript text (including legends for main figures, EV figures and tables). Please make sure that the changes are highlighted to be clearly visible.
- 2) Individual production quality figure files as .eps, .tif, .jpg (one file per figure). For guidance, download the 'Figure Guide PDF': (<https://www.embopress.org/page/journal/17574684/authorguide#figureformat>).
- 3) A .docx formatted letter INCLUDING the reviewers' reports and your detailed point-by-point responses to their comments. As part of the EMBO Press transparent editorial process, the point-by-point response is part of the Review Process File (RPF), which will be published alongside your paper.
- 4) A complete author checklist, which you can download from our author guidelines (<https://www.embopress.org/page/journal/17574684/authorguide#submissionofrevisions>). Please insert information in the checklist that is also reflected in the manuscript. The completed author checklist will also be part of the RPF.
- 5) Please note that all corresponding authors are required to supply an ORCID ID for their name upon submission of a revised

manuscript.

6) It is mandatory to include a 'Data Availability' section after the Materials and Methods. Before submitting your revision, primary datasets produced in this study need to be deposited in an appropriate public database, and the accession numbers and database listed under 'Data Availability'. Please remember to provide a reviewer password if the datasets are not yet public (see <https://www.embopress.org/page/journal/17574684/authorguide#dataavailability>).

13) Author contributions: You will be asked to provide CRediT (Contributor Role Taxonomy) terms in the submission system. These replace a narrative author contribution section in the manuscript.

14) A Conflict of Interest statement should be provided in the main text.

Please also suggest a striking image or visual abstract to illustrate your article as a PNG file 550 px wide x 300-800 px high.

***** Reviewer's comments *****

Referee #1 (Comments on Novelty/Model System for Author):

The study describes novel findings and how a antidiabetic drug that us widely used by a large proportion of the population, could affect immune responses to vaccines. This information has wide ranging implications in adult vaccination programs.

Referee #1 (Remarks for Author):

The manuscript by Mok et al, highlights some very important findings and sheds light on how drugs such as metformin could alter immune responses to vaccines. The authors have carefully conducted the study and presented their data. I have the following questions/comments.

Results:

1. Study subjects on metformin: as these were healthy subjects, did any of them have hypoglycaemic episodes?
2. The authors show that the systemic side effects were significantly less (19.2%) in this study vs previous studies: could these changes also be attributed to a lower dose used in the YFV in this study, the age groups being younger or the sample size being significantly smaller than previous studies?
3. Viraemia was only detected in 76.9%. Were systemic side effects present in the aviraemic or the viraemic group?
4. The most interesting finding of this study is the metformin treated group having higher Neut50 titres despite some being aviraemic. The gene transcriptomics show that type IFN responses and inflammatory responses were both upregulated in those on metformin treatment. What is the authors opinion on upregulation of both antiviral and inflammatory responses in those with metformin?
5. The authors had looked at plasma metabolites which show significant differences. Have they had the opportunity to look at SCFAs, which may associate with a more robust B cell response and antigen presentation? Also reducing inflammation?
6. Could any of these differences due to those on metformin treatment be attributed to changes in the gut microbiome with metformin? Hagan et al have shown altered responses to the IAV following antibiotics, which were due to changes in the gut microbiome.

Referee #2 (Comments on Novelty/Model System for Author):

Could be corroborated by confirming hypothesis from human trial in small animal model.

Referee #2 (Remarks for Author):

Mom and colleagues describe a human trial investigating the impact of metformin (anti-diabetic drug) treatment on live-attenuated YF17D immunisation with intriguing and thought-provoking findings. Though the final study groups are small, careful data mining seem to unveil relevant biology.

In my opinion the authors should be more careful with two things. Firstly, comparing live-attenuated vaccines (i.e. YF17D) with inactivated vaccines (i.e. seasonal flu) when seeking for comparators from literature. Second, use of the term "correlate (correlation) of immunization". Please see also my specific comments in the following.

A key hypothesis derived from their clinical observation, enhancement of YF17D vaccination by a brief course of metformin

should be fairly easy to be confirmed in small animal models. The discussion leaves me honestly with some confusion. While in the actual study drug treatment enhanced immunization, exactly the opposite is elaborated in the last long paragraph of the Discussion.

Specific comments and questions.

Abstract

Line 29: The authors try to introduce clear definitions for correlates of vaccine induced immunity. This much appreciated yet their choice of wording is somewhat uncommon. Correlates are by definition mathematical statistical. By contrast, "causation" or "association" may be more appropriate for describing a functional/mechanistic link. I suggest to avoid the new wording "functional correlate".

Line 31: please identify metformin as anti-diabetic drug.

Line 40: This concluding sentence suggests that YF17D (or other live-attenuated) may not be efficacious in diabetic patients. What is the clinical evidence for this? Caution should be taken not to mix statements relating to vaccines with different mechanism of action (e.g. live vs. inactivated viral vaccines).

Line 89: I suggest to specify final study group by additional wording "... of the volunteers (n=26)"

Line 94 referring to Suppl. Fig 2B: Please specify in the Figure legend which reference refers to which Trial 1 or 2; reference Chan et al. JCI (2017) incomplete; please list and specify "systemic symptoms"

Line 123: Please avoid the term "direct correlation" as it is misleadingly suggesting a mechanistic link.

Line 125, 246 and 263: peri-vaccination is not a generally used term. Please try to be more precise what is meant here.

Line 129: Please be careful with introducing the term "association" here which is frequently used for a "mechanistic" link or causation (in contrast to mathematical "correlation" that may be meant here)

Line 136-145: Data shown in Fig. S4A,C may suggest a profound effect of metformin on longevity of T cell responses as measured by ELISpot. Though based on a very low number of samples (n=4-6), I feel this needs to be critically discussed as a possible risk of metformin treatment as long-term YF immunity is also based on cellular memory responses.

Line 147-157: This statement is somewhat in conflict with the aforementioned up-regulation of innate anti-viral responses at pre-vaccination (D0) (Fig. S3B). What happened to the profile of those with high innate tone at D0, no viremia and low PRNT50 titers. Do these individuals still show an elevation of cytokines over time? See also Muyenja JCI 2017 PMID: 24911151 observing the same phenomenon, however also interference with cell-mediated immunity. Can these conflicting data be discussed?

Line 163: How were these each n=5 subjects per group selected?

Line 174: abbreviation ETC to be introduced

Line 183 and Fig3: How are "leading edge genes" defined by the authors?

Line 186: The authors speculate that OXPHOS is particularly playing a role in B cell response to metformin treated YF17D vaccinees, also because they do not see a T cell signal in their ELISpot assessment. scRNAseq would solve this debate/support this hypothesis which is obviously not possible due to the study protocol. Can this shortcoming at least be discussed carefully.

Line 191-192 related to Fig. 3D: It is not clear how "elevated" is meant here. Elevated relatively data from which sampling time point, D-3 or D0. Statistical significance should be revisited accordingly (not only comparing means from d3 vs D7 and D7 vs. D10). More generally, early YF17D humoral response should be dominated by IgM. What is the evidence that the IgG expression is related to specific responses and not general immune activation (bystander effect)?

Line 197: D3-D7 is considered a relevant time point for the development of YF-specific plasma blasts. This time point is based on literature of LPS (T-cell independent) stimulation of B cells in mice. By contrast, YF17D is known to induce very rapidly (in less than 5 days) high levels of nAb (mainly IgM but also already IgG) in humans. Are these timepoints then well chosen. Can the authors also provide kinetic information regarding seroconversion in both study groups? Now for all considerations only 28d data (at equilibrium) are shown. Likewise nAb could peak much earlier. Seroconversion kinetics and/or AUC for nAb may provide a higher resolution to possibly show better separation of groups. Also, a more direct link could be made with metabolic changes.

Fig. 4: For me it is not clear why the authors do not compare also d3 and d7 values between placebo and treatment group. For some metabolites and DEG there seems to be a marked difference (e.g. succinate or lactate at D7, or GSS at D3). Thus why

only pairwise analysis within groups? Are the groups too small for power calculation?

Line 233: please spell T2DM out

Line 250: I may have misunderstood, but suddenly the discussion goes in a direction as if metformin treatment was not enhancing B-cell responses (in contrast to data shown in Fig. 1D). Why should metformin have an adverse effect on (YF17D) vaccine efficacy? Literature cited here is referring to T-cell independent B-cell activation, or vaccination using inactivated whole virus antigens. Regarding YF17D, are the authors suggesting that their treatment schedule lasted for too long? Would they thus expect an even more pronounced effect of a short pulse of metformin. This is an easily testable hypothesis that could be assessed in short term animal experiments.

Line 257: YF17D vaccination and immunology immunization using inactivated flu vaccine can hardly be compared (see e.g. Nakaya et al. Nat Immunol. 2011 PMID: 21743478; Li et al. Nat Immunol. 2014 PMID: 24336226) Mechanistically they need rather to be considered two extremes in a spectrum of types of antiviral vaccines.

Lines 266: Please see my comment on line 29. Mitochondrial activity seems thus to play a role in yellow fever immunization, whereas the organelle "mitochondria" (as such) correlated neither statistically nor "functionally" with the induction of of nAb.

Lines 277: Please specify Stamaril lot number and expiry date, also considering the drop in titer compared to the other two founding studies shown in Fig. S2.

Line 278-279: Samples should be available to do antibody kinetics, see our comment to line 197.

Line 312-315: qRT-PCR was key to score for viremia (Fig. 1+2). How was LOD determined? What was used as standard to convert Ct values to mRNA copies? What the level of validation of this in house assay?

Line 457-459: Please see my comment on line 29. Please be careful with shuffling terms "correlation" and "association" and deliberately combining with "statistical" and "functional"

Line 471-472: Item

The study has been conducted in a dengue endemic area. Can a potential of flavivirus cross-reactivity and/or enhancement of YF17D immunogenicity in dengue (flavivirus) seropositive subjects be ruled out. In Table S1 DENV IgG aerostats in mentioned (as all negative). How was this measured? What was the YF serostatus prior to the study, also considering that PRNT50 values are used for all correlation analysis in Fig. 1-2 and S8-9. Would there be another outcome if PRNT80 or 90 values (with resulting height specificity) were used for correlation analysis?

Referee #3 (Comments on Novelty/Model System for Author):

The study, employing a randomized double-blind placebo-controlled trial format, offers significant insights into YF17D vaccine immunogenicity. A total of 32 healthy volunteers were recruited and administered either a placebo or metformin treatment before and after YF17D vaccination. The bioinformatics and statistical analyses were comprehensive and well-executed. The methods for viremia assessment, immunological testing, and transcriptomic profiling are detailed, adhering to current scientific standards. The finding that metformin treatment enhances the immunogenicity of the YF17D vaccine in volunteers with low yellow fever 17D viremia is interesting. This could significantly impact strategies for improving vaccine efficacy, particularly in populations with altered metabolic states. However (and this is a big "however"), the study's reliance on a very small sample size is a significant limitation, potentially affecting the robustness and reliability of the findings. A larger sample might provide more robust results. Additionally, focusing solely on healthy volunteers may limit the generalizability of the findings to broader populations, especially those with comorbidities commonly treated with metformin, such as diabetes.

Referee #3 (Remarks for Author):

My major and sole concern is the small sample size, which could potentially limit the robustness and generalizability of the study's findings.

Please see below for a point-by-point response to the issues raised by the referees:

Referee #1 (Comments on Novelty/Model System for Author):

The study describes novel findings and how a antidiabetic drug that us widely used by a large proportion of the population, could affect immune responses to vaccines. This information has wide ranging implications in adult vaccination programs.

We thank Referee #1 for this supportive remark.

Referee #1 (Remarks for Author):

The manuscript by Mok et al, highlights some very important findings and sheds light on how drugs such as metformin could alter immune responses to vaccines. The authors have carefully conducted the study and presented their data. I have the following questions/comments.

Results:

1. Study subjects on metformin: as these were healthy subjects, did any of them have hypoglycaemic episodes?

Although metformin is widely used as an anti-diabetic agent, metformin monotherapy, due to its mode of action on hepatic gluconeogenesis, is rarely associated with hypoglycemia (UK Prospective Diabetes Study Group, 1995; UK Prospective Diabetes Study Group). None of the subjects in our trial experienced any hypoglycemic episodes.

2. The authors show that the systemic side effects were significantly less (19.2%) in this study vs previous studies: could these changes also be attributed to a lower dose used in the YFV in this study, the age groups being younger or the sample size being significantly smaller than previous studies?

We thank Referee #1 for this question. The lower rate of systemic side effects was, indeed, more likely due to the lower dose of YF17D in the batch of vaccines supplied at the time of this study (Lines 98 to 103). It is unlikely that age of the study participants was a confounder, as the age groups of subjects in this trial [Mean (SD) = 32y (4.9)] were similar to that of our previous study [Chan et al, 2016; Mean (SD) = 31.8y (7.9)].

3. Viraemia was only detected in 76.9%. Were systemic side effects present in the aviraemic or the viraemic group?

Referee #1 brings up an interesting point. We did not observe any systemic side effects in the aviremic subjects, and the majority of viremic subjects also did not

report any systemic side effects. Only five viremic subjects (three in the placebo group and two in the metformin group) reported experiencing systemic side effects. Furthermore, among the viremic subjects, we found no significant difference in viremia levels between those who reported systemic side effects and those who did not. We have not included these findings in our revised manuscript as they are consistent with our earlier study, which showed no association between viremia and systemic side effects (Chan et al, 2017).

Comparison of viremia (AUC) levels between viremic subjects who were asymptomatic or symptomatic following YF17D vaccination. Circles and triangles represent subjects in the Placebo and Metformin groups respectively.

- The most interesting finding of this study is the metformin treated group having higher Neut50 titres despite some being aviraemic. The gene transcriptomics show that type IFN responses and inflammatory responses were both upregulated in those on metformin treatment. What is the authors opinion on upregulation of both antiviral and inflammatory responses in those with metformin?

We thank Referee #1 for this interesting question. We do not know the exact reason for the observed trends in IFN and inflammatory responses. They could be due to the higher frequency of circulating peripheral myeloid dendritic cells in the metformin group compared to the placebo as shown in Appendix Fig S6E. Indeed, it has been previously shown in healthy adult volunteers that metformin treatment modulates the circulating immune cell landscape and increases the proportion of mononuclear myeloid cells in PBMCs (Lachmandas et al, 2019). Future studies will be needed to dissect how metformin leads to changes in expression of genes in these pathways.

- The authors had looked at plasma metabolites which show significant differences. Have they had the opportunity to look at SCFAs, which may associate with a more robust B cell response and antigen presentation? Also reducing inflammation?

We thank Referee #1 for this question. We have focused on polar metabolites, given the known function of metformin in glucose metabolism. Lipid metabolites may

indeed be involved in inflammation, which could indirectly affect B cell response to YF17D. We have now added this limitation into our discussion (Lines 276 to 285).

6. Could any of these differences due to those on metformin treatment be attributed to changes in the gut microbiome with metformin? Hagan et al have shown altered responses to the IAV following antibiotics, which were due to changes in the gut microbiome.

We thank Referee #1 for this insightful question. The gut microbiome can indeed influence adaptive immune function, and we acknowledge that we should have considered this possibility in our original manuscript. However, the addition of new mouse data, as described in Lines 225 to 245 and shown in Figure 5, argues against this possibility. If indeed, metformin induced microbiome changes explain the immunological outcome of our study, there should be no difference in antibody response regardless of when treatment was stopped. Figure 5 shows that enhanced antibody response was observed in animals that received metformin which was stopped at D3 post-vaccination, similar to our study volunteers. In contrast, continuation of metformin treatment till D10 post-vaccination did not result in similar enhancement in YF17D antibody response. These findings thus support a more direct role for early OXPHOS responses in shaping vaccine-induced humoral immunity. We have now added this discussion into Lines 282 to 285.

Referee #2 (Comments on Novelty/Model System for Author):

Could be corroborated by confirming hypothesis from human trial in small animal model.

Thank you for this excellent suggestion. We have conducted a mouse study to confirm our hypothesis. This data has been added in our revised manuscript (further elaboration below).

Referee #2 (Remarks for Author):

Mok and colleagues describe a human trial investigating the impact of metformin (anti-diabetic drug) treatment on live-attenuated YF17D immunisation with intriguing and thought-provoking findings. Though the final study groups are small, careful data mining seem to unveil relevant biology.

In my opinion the authors should be more careful with two things.

1. Firstly, comparing live-attenuated vaccines (i.e. YF17D) with inactivated vaccines (i.e. seasonal flu) when seeking for comparators from literature. Second, use of the term "correlate (correlation) of immunization". Please see also my specific comments in the following.

We thank Referee #2 for questioning the precision of our description. We have made the necessary amendments along with others suggested in Questions 3 and 20.

2. A key hypothesis derived from their clinical observation, enhancement of YF17D vaccination by a brief course of metformin should be fairly easy to be confirmed in small animal models. The discussion leaves me honestly with some confusion. While in the actual study drug treatment enhanced immunization, exactly the opposite is elaborated in the last long paragraph of the Discussion.

We thank Referee #2 for this excellent suggestion! We have now conducted the additional experiment. We chose C57BL/6 mice as this mouse strain is known to be refractory to YF17D viremic infection, thus simulating the subjects with no and low YF17D viremia in our clinical study that showed augmented antibody response with short course metformin. We are delighted to report that the mouse experiment fully supported our clinical observation. This data is now presented in Figure 5. We have also revised our discussion accordingly in Lines 298 to 311.

Specific comments and questions.

1. Abstract

Line 29: The authors try to introduce clear definitions for correlates of vaccine induced immunity. This much appreciated yet their choice of wording is somewhat uncommon. Correlates are by definition mathematical statistical. By contrast, "causation" or "association" may be more appropriate for describing a functional/mechanistic link. I suggest to avoid the new wording "functional correlate".

We agree with Referee #2 that the wording can be better structured to clearly define the difference between “correlation” and association”. Hence, for increased clarity, we have now removed the words “functional correlate” and have replaced it with “association” to indicate any potential functional or mechanistic links throughout the text.

2. Line 31: please identify metformin as anti-diabetic drug.

We have added the words “anti-diabetic drug” to identify metformin as such (Line 32 to 33).

3. Line 40: This concluding sentence suggests that YF17D (or other live-attenuated) may not be efficacious in diabetic patients. What is the clinical evidence for this? Caution should be taken not to mix statements relating to vaccines with different mechanism of action (e.g. live vs. inactivated viral vaccines).

We thank Referee #2 for raising this important point and acknowledge the need for caution in mixing statements relating to vaccines with different mechanisms of action. We have thus reworded our concluding sentence to “Our findings thus demonstrate a functional association between cellular respiration and vaccine-induced humoral immunity and suggest potential approaches to enhancing vaccine immunogenicity.” (Lines 41 to 43).

4. Line 89: I suggest to specify final study group by additional wording "... of the volunteers <enrolled in final analysis> (n=26 <of which n=12 placebo and n=14 metformin treated>)"

We have formatted the sentence to include the suggested text (Lines 91 to 93).

5. Line 94 referring to Suppl. Fig 2B: Please specify in the Figure legend which reference refers to which Trial 1 or 2; reference Chan et al. JCI (2017) incomplete; please list and specify "systemic symptoms"

We apologize for the incompleteness of our references. We have revised the figure legend for Appendix Fig S2 to specify which references pertain to Trial 1 and Trial 2. The term “systemic symptoms” were based on the Common Terminology Criteria for Adverse Events (CTCAE) definitions and are divided into organ systems: central nervous system (CNS), musculoskeletal (MSK), gastrointestinal (GI), or respiratory symptoms during analysis. These details are now described in Lines 96 to 98.

6. Line 123: Please avoid the term "direct correlation" as it is misleadingly suggesting a mechanistic link.

We have removed the term “direct correlation” and replaced it with “known relationship” (Line 129).

7. Line 125, 246 and 263: peri-vaccination is not a generally used term. Please try to be more precise what is meant here.

We have removed the term “peri-vaccination” throughout the text and replaced it with “a brief course of metformin treatment, given 3-days before, to 3-days after YF17D vaccination.”

8. Line 129: Please be careful with introducing the term "association" here which is frequent used for a "mechanistic" link or causation (in contrast to mathematical "correlation" that may be meant here)

We have replaced the word “association” with “correlation” (Line 135).

9. Line 136-145: Data shown the Fig. S4A,C may suggest a profound effect of metformin on longevity of T cell responses as measured by ELISpot. Though based on a very low number of samples (n=4-6), I feel this needs to be critically discussed as a possible risk of metformin treatment as long term YF immunity is also based on cellular memory responses.

We thank Referee #2 for this insightful comment. Unfortunately, our trial was not designed to examine the longevity of T cell responses. Nevertheless, we have now included this consideration into our discussion (Lines 286 to 297).

10. Line 147-157: This statement is somewhat in conflict with the aforementioned up-regulation of innate anti-viral responses at pre-vaccination (D0) (Fig. S3B). What happened to the profile of those with high innate tone at D0, no viremia and low PRNT50 titers. Do these individuals still show an elevation of cytokines over time? See also Muyanja JCI 2017 PMID: 24911151 observing the same phenomenon, however also interference with cell-mediated immunity. Can these conflicting data be discussed?

We thank Referee #2 for the question and apologize for any confusion. To provide further clarification, in Fig. S3B, we examined the D0 pre-vaccination host response, and found that the six aviremic individuals exhibited elevated innate anti-viral responses compared to their viremic counterparts. Among the aviremic subjects, three of them were from the placebo group, while the remaining three were from the metformin group. Therefore, the reference to subjects with high innate tone at D0, no viremia, and low PRNT50 titers pertains to the three aviremic subjects from the placebo group. For clarity, we have now removed Fig 1C and moved Fig 1D in its place, and revised our manuscript text accordingly (Lines 122 to 130).

Regarding cytokine and cell-mediated responses, we have re-analyzed the data, specifically comparing the responses between aviremic and viremic subjects within

the placebo arm. We found no differences in T-cell IFN γ ELISpot responses at all sampled timepoints. Furthermore, we found lower levels of IL27 at D3 post-vaccination in aviremic subjects compared to viremic subjects, while the expression of all other cytokines was not significantly different at any sampled timepoints. Overall, our data suggests that the reduction in viremia is unlikely to be attributed to differences in cell-mediated immunity or cytokine levels.

Comparison of T-cell responses and cytokine concentrations between placebo-treated subjects with undetectable or detectable YF17D viremia (A) Total ex-vivo T-cell IFN γ ELISpot responses against the complete YF17D proteome at D0 (before vaccination), D14 and D28 post-vaccination in placebo-treated subjects with or without detectable viremia. (B) Volcano plot illustrating the difference in log₂ mean concentration of 37 cytokines between placebo-treated subjects with or without detectable viremia at -D3 (pre-treatment baseline), D0 (before vaccination), D3, D7 and D10 post-vaccination.

11. Line 163: How were these each n=5 subjects per group selected?

There was a total of 11 subjects with low or no detectable viremia. However, only ten were included for analysis as samples for one subject were not available at all the timepoints. We have thus reworded the manuscript for additional clarity (Lines 162 to 165).

12. Line 174: abbreviation ETC to be introduced

We have added the words “electron transport chain” before the acronym “ETC” (Lines 174 to 175).

13. Line 183 and Fig3: How are "leading edge genes" defined by the authors?

We defined “leading edge genes” based on the definition provided by Subramanian et al (2005), which referred to these genes as “the genes of a gene set that account for the enrichment signal.” To clarify this, we have updated our Materials and Methods to include the definition of a leading edge gene (Lines 404 to 406).

14. Line 186: The authors speculate that OXPHOS is particularly playing a role in B cell response to metformin treated YF17D vaccinees, also because they do not see a T cell signal in their ELISpot assessment. scRNAseq would solve this debate/support this hypothesis which is obviously not possible due to the study protocol. Can this shortcoming at least be discussed carefully.

We thank Referee #2 for raising this point and agree that single cell assays would have provided additional insights into the enhancing effect of metformin on B cell, but not T cell responses in our study. While we are not able to examine this due to limitation of study protocol and consent, we have nevertheless added this as a discussion point in our main text (Line 286 to 297).

15. Line 191-192 related to Fig. 3D: It is not clear how "elevated" is meant here. Elevated relatively data from which sampling time point, D-3 or D0. Statistical significance should be revisited accordingly (not only comparing means from d3 vs D7 and D7 vs. D10). More generally, early YF17D humeral response should be dominated by IgM. What is the evidence that the IgG expression is related to specific responses and not general immune activation (bystander effect)?

We thank Referee #2 for highlighting this confusion. The elevation described is relative to -D3 (pre-treatment). This information has been added to the text (Lines 183 to 185). The data in Fig. 3D was derived by calculating the mean \log_2 expression counts of immunoglobulin genes, including those related to IgM and IgG, for each subject. The list of immunoglobulin genes analyzed are listed in Appendix Table S3. Regarding the specificity of the response, we had measured the levels of neutralizing antibodies as an endpoint, which is a specific response. While we are not able to discount a bystander effect as we did not measure total IgG levels, we are confident that the IgG expression is related to specific responses.

16. Line 197: D3-D7 is considered a relevant time point for the development of YF-specific plasma blasts. This time point is based on literature of LPS (T-cell independent) stimulation of B cells in mice. By contrast, YF17D is known to induce very rapidly (in less than 5 days) high levels of nAb (mainly IgM but also already IgG) in humans. Are these timepoints then well chosen. Can the authors also provide kinetic information regarding seroconversion in both study groups? Now for all considerations only 28d data (at equilibrium) are shown. Likewise nAb could peak much earlier. Seroconversion kinetics and/or AUC for nAb may provide a higher resolution to possibly show better separation of groups. Also, a more direct link could be made with metabolic changes.

While YF17D is known to induce IgM antibodies very rapidly, studies have shown that the peak neutralizing antibody response occurs at D28 post-vaccination (Barret and Teuwen, 2009; Niedrig et al, 2002; Wec et al, 2019). The difference we observed in the neutralizing antibody response to YF17D vaccination is the titer and not the kinetics of antibody expression. Hence, we focused our analysis on the

RNAseq data for genes and pathways that correlate with YF17D PRNT₅₀ titers. The effects of metformin on the kinetics of B cell response to YF17D vaccination will need to be a subject for future investigations and we have included this as a limitation of our study (Lines 293 to 295).

17. Fig. 4: For me it is not clear why the authors do not compare also d3 and d7 values between placebo and treatment group. For some metabolites and DEG there seems to be a marked difference (e.g. succinate or lactate at D7, or GSS at D3). Thus why only pairwise analysis within groups? Are the groups too small for power calculation?

We thank Referee #2 for the question. Our findings were presented to us by chance; we had not set out to test the effect of metformin on B metabolism following YF17D infection. The sample size that we had to work with to understand the difference in outcome based on neutralizing antibody titers is small and insufficiently powered for us to compare between the two groups. To glean insights into how increased expression of genes in the ETC supported improved PRNT₅₀ titers, we thus performed pairwise analysis within the groups as we are examining the changes in host response within each subgroup following the cessation of drug therapy. For better clarity, we have now rephrased the text to emphasize that “the functional effects of the increased expression of ETC-related genes could be gleaned from the time-dependent changes plasma polar metabolites in the subgroups” (Lines 196 to 197).

18. Line 233: please spell T2DM out

We have added the words “Type 2 Diabetes Mellitus” before the acronym “T2DM” (Lines 260).

19. Line 250: I may have misunderstood, but suddenly the discussion goes in a direction as if metformin treatment was not enhancing B-cell responses (in contrast to data shown in Fig. 1D). Why should metformin have an adverse effect on (YF17D) vaccine efficacy? Literature cited here is referring to T-cell independent B-cell activation, or vaccination using inactivated whole virus antigens. Regarding YF17D, are the authors suggesting that their treatment schedule lasted for too long? Would they thus expect an even more pronounced effect of a short pulse of metformin. This is an easily testable hypothesis that could be assessed in short term animal experiments.

We apologize that the wordings may have misled the reviewer. For clarity, we have rephrased discussion lines 251 to 254 to indicate that it is the brief course of metformin treatment, given 3-days before, to 3-days after YF17D vaccination, that resulted in the enhanced humoral response observed. When metformin was stopped at D3 post-vaccination, which coincided with the known timepoint where B cells upregulated OXPHOS following stimulation, the release of inhibition by metformin on the accumulated expression of respiratory complex transcripts in B cells at this point could have thus supported more efficient OXPHOS and hence humoral responses.

In addition, to demonstrate the importance of metformin treatment and cessation at D3 post YF17D vaccination on neutralizing antibody response, we have now performed additional experiments in a C57BL/6 mouse model. This mouse strain was selected due to its well-documented inability to support viremic YF17D infection (Erickson & Pfeiffer, 2015), which thus mimics our clinical trial subjects with low or no viremia. We found that compared to untreated mice, metformin treatment resulted in significantly higher neutralizing antibody titers and seroconversion rates when metformin treatment was discontinued at D3 but not at D10 post YF17D vaccination. These data have now been added in Lines 225 to 245 and presented in Figure 5. Overall, our data thus support the notion that treatment and timing of metformin cessation may play a critical role in vaccination outcomes.

20. Line 257: YF17D vaccination and immunology immunization using inactivated flu vaccine can hardly be compared (see e.g. Nakaya et al. Nat Immunol. 2011 PMID: 21743478; Li et al. Nat Immunol. 2014 PMID: 24336226) Mechanistically they need rather to be considered two extremes in a spectrum of types of antiviral vaccines.

We thank Referee #2 for highlighting this point and acknowledge the importance of caution when comparing different vaccine types. However, the poor immunologic response to vaccination observed in diabetic patients also extends to other live-attenuated vaccines. Indeed, a retrospective study by Bollaerts et al (2019) demonstrated decreased vaccine effectiveness of the live-attenuated Varicella-Zoster Virus vaccine (Zostavax) against shingles in elderly patients with type 2 diabetes compared to their non-diabetic counterparts. While the role of metformin in this observation requires further investigation, our findings in YF17D vaccinated mice highlight the potential negative impact of extended metformin therapy on live-attenuated vaccination responses. For greater clarity, we have now revised our manuscript to discuss this point in greater detail (Lines 298 to 311).

21. Lines 266: Please see my comment on line 29. Mitochondrial activity seems thus to play a role in yellow fever immunization, whereas the organelle "mitochondria" (as such) correlated neither statistically nor "functionally" with the induction of nAb.

We thank the Referee for highlighting this, and we have formatted the text to indicate that it is mitochondrial respiration that is functionally associated with antibody response to live-attenuated YF17D vaccination (Lines 312 to 313).

22. Lines 277: Please specify Stamaril lot number and expiry date, also considering the drop in titer compared to the other two founding studies shown in Fig. S2.

We have updated the text to include the Stamaril lot number and expiry date (Lines 322 to 323).

23. Line 278-279: Samples should be available to do antibody kinetics, see our comment to line 197.

Antibody kinetics is certainly important for pandemic response, where earlier acquisition of the correlates of protection may be particularly beneficial. Respectfully, however, this is not the focus of this study and will have to be a subject for future investigations.

24. Line 312-315: qRT-PCR was key to score for viremia (Fig. 1+2). How was LOD determined? What was used as standard to convert Ct values to mRNA copies? What the level of validation of this in house assay?

We used a cycle threshold value of 40 as the limit of detection. The standards were derived from YF17D RNA standards that were quantified with a known copy number per mL. Technical duplicates were carried out for each standard concentration, and the values of eight independent runs were consolidated to construct the standard curve used to convert Ct values to mRNA copies. We have now added this information in the Materials and Methods (Lines 357 to 363).

25. Line 457-459: Please see my comment on line 29. Please be careful with shuffling terms "correlation" and "association" and deliberately combining with "statistical" and "functional"

We have reworded the manuscript to clearly define the use of "statistical correlates" and "associations".

26. Line 471-472: The study has been conducted in a dengue endemic area. Can a potential of flavivirus cross-reactivity and/or enhancement of YF17D immunogenicity in dengue (flavivirus) seropositive subjects be ruled out. In Table S1 DENV IgG aerostats is mentioned (as all negative). How was this measured? What was the YF serostatus prior to the study, also considering that PRNT50 values are used for all correlation analysis in Fig. 1-2 and S8-9. Would there be another outcome if PRNT80 or 90 values (with resulting height specificity) were used for correlation analysis?

We thank the Referee for the comments. The DENV IgG serostatus was ascertained prior to enrollment using the commercial Panbio Dengue IgG Indirect ELISA kit (Line 319). All subjects were pre-screened for YF serostatus, and only those who have declared to have never received YF17D vaccination before were eligible for enrollment. We have revised our Materials and Methods to reflect these criteria (Line 318). Regarding the choice of PRNT50 values for correlation analyses, we employed this titer because it is the most widely used measure of neutralizing antibody responses. We have also re-analyzed the data using PRNT80 and PRNT90 values and found that the same subjects were classified into the same categories (no viremia, low viremia, and viremia) regardless of the PRNT titer used. Hence, we do

not anticipate a different outcome if PRNT80 or PRNT90 values are used for analyses.

Comparison of PRNT80 titers between the Placebo and Metformin groups. (A) \log_2 PRNT80 titers of subjects with or without detectable viremia. Subjects in the placebo and metformin groups are colored black and gold respectively. (B) Comparison of \log_2 PRNT80 titers by placebo and metformin groups. Relationship between \log_2 PRNT80 titers and viremia (AUC) in (C) placebo treated and (D) metformin treated subjects. r values indicate Pearson's correlation between PRNT80 titers and viremia (AUC) with the corresponding p -values. Statistical analyses were performed using unpaired Student's t -test, where $*p < 0.05$.

Comparison of PRNT90 titers between the Placebo and Metformin groups. (A) Log₂ PRNT80 titers of subjects with or without detectable viremia. Subjects in the placebo and metformin groups are colored black and gold respectively. (B) Comparison of Log₂ PRNT90 titers by placebo and metformin groups. Relationship between Log₂ PRNT90 titers and viremia (AUC) in (C) placebo treated and (D) metformin treated subjects. r values indicate Pearson's correlation between PRNT90 titers and viremia (AUC) with the corresponding p -values. Statistical analyses were performed using unpaired Student's t -test, where $*p < 0.05$.

Referee #3 (Comments on Novelty/Model System for Author):

The study, employing a randomized double-blind placebo-controlled trial format, offers significant insights into YF17D vaccine immunogenicity. A total of 32 healthy volunteers were recruited and administered either a placebo or metformin treatment before and after YF17D vaccination. The bioinformatics and statistical analyses were comprehensive and well-executed. The methods for viremia assessment, immunological testing, and transcriptomic profiling are detailed, adhering to current scientific standards. The finding that metformin treatment enhances the immunogenicity of the YF17D vaccine in volunteers with low yellow fever 17D viremia is interesting. This could significantly impact strategies for improving vaccine efficacy, particularly in populations with altered metabolic states. However (and this is a big "however"), the study's reliance on a very small sample size is a significant limitation, potentially affecting the robustness and reliability of the findings. A larger sample might provide more robust results. Additionally, focusing solely on healthy volunteers may limit the generalizability of the findings to broader populations, especially those with comorbidities commonly treated with metformin, such as diabetes.

We thank Referee #3 for the positive impression and agree with the need for validation through a larger study (See response to other questions for details).

Referee #3 (Remarks for Author):

1. My major and sole concern is the small sample size, which could potentially limit the robustness and generalizability of the study's findings.

We acknowledge that the small sample size and focus on healthy volunteers are limitations of our findings. To address these concerns, we have now validated our findings using a small animal model, and demonstrated that compared to untreated C57BL/6 mice, mice treated with metformin until D3, but not D10, post-YF17D vaccination exhibited enhanced neutralizing antibody titers and seroconversion rates (Lines 225 to 245). The data for this experiment is now presented in Figure 5. However, we agree with the reviewer that future work including examining how metformin modulates vaccine immunogenicity in patients on long-term metformin therapy such as type 2 diabetic patients would need to be carried out in order to determine the generalizability and applicability of these findings in clinical settings. We have now included a discussion regarding the clinical implications of our findings in Lines 298 to 311.

22nd Mar 2024

Dear Dr. Low,

Thank you for the submission of your revised manuscript to EMBO Molecular Medicine. I am pleased to inform you that we will be able to accept your manuscript pending the following final amendments:

- 1) Please address the minor point raised by the referee.
- 2) Authors: Please make sure that the authors names are the same in our submission system and in the manuscript. Majority of authors have either a middle name or middle name initials in the manuscript, while these are missing in our system. For example, Lavanya Lakshmi Jeeva in the manuscript file vs. Lavanya Jeeva in our system.
- 3) In the main manuscript file, please do the following:

- Please address all comments suggested by our data editors listed below:

o Figure legends:

1. Please note that the box plots need to be defined in terms of minima, maxima, centre, bounds of box and whiskers, and percentile in the legends of figures 4a-b; 5c-d.
2. Please note that information related to n is missing in the legends of figures 1b-c; 2d.
3. Please note that the error bars are not defined in the legends of figures 5c-d.

- In M&M, provide the statement that the experiments involving human participants conformed to the principles set out in the WMA Declaration of Helsinki and the Department of Health and Human Services Belmont Report. Please check "Author Guidelines".

<https://www.embopress.org/page/journal/17574684/authorguide#humansubjects>

- For clinical trials reporting, the authors should fill out a CONSORT flow diagram and submit it as supplementary information. The journal also encourages authors to follow the CONSORT reporting guidelines <http://www.consort-statement.org>. Please see the EQUATOR website for details. Clinical trials should also be registered as recommended by the International Committee of Medical Journal Editors, and the trial registration number should be provided. Please check "Author Guidelines".

<https://www.embopress.org/page/journal/17574684/authorguide#humansubjects>

- In M&M, please specify the biosafety level for the experiments with YF17D virus by adding and amending the following sentence: All experiments with YF17D virus were performed in a ... level laboratory and with approval from...

- In M&M, statistical paragraph should reflect all information that you have filled in the Authors Checklist, especially regarding randomization, blinding, replication.

- Author contributions: Please remove it from the manuscript and specify author contributions in our submission system. CRediT has replaced the traditional author contributions section because it offers a systematic machine-readable author contributions format that allows for more effective research assessment. You are encouraged to use the free text boxes beneath each contributing author's name to add specific details on the author's contribution. More information is available in our guide to authors:

<https://www.embopress.org/page/journal/17574684/authorguide#authorshipguidelines>

- Data availability: Please use the following format to report the accession number of your data:

[data type]: [full name of the resource] [accession number/identifier] [(doi or URL or identifiers.org/DATABASE:ACCESSION)]

Please check "Author Guidelines" for more information.

<https://www.embopress.org/page/journal/17574684/authorguide#availabilityofpublishedmaterial>

4) Appendix: Please add page numbers in the table of content and remove zip folder with appendix figures.

5) Synopsis:

- Synopsis image: Please resize the image to 550 px-wide x (250-400)-px high and upload it as a high-resolution jpeg file.

6) Source data: Please upload one file per figure.

7) For more information: This space should be used to list relevant web links for further consultation by our readers. Could you identify some relevant ones and provide such information as well? Some examples are patient associations, relevant databases, OMIM/proteins/genes links, author's websites, etc...

8) As part of the EMBO Publications transparent editorial process initiative (see our Editorial at

<http://embomolmed.embopress.org/content/2/9/329>), EMBO Molecular Medicine will publish online a Review Process File (RPF) to accompany accepted manuscripts. This file will be published in conjunction with your paper and will include the anonymous referee reports, your point-by-point response and all pertinent correspondence relating to the manuscript. Let us know whether you agree with the publication of the RPF and as here, if you want to remove or not any figures from it prior to publication. Please note that the Authors checklist will be published at the end of the RPF.

9) Please provide a point-by-point letter INCLUDING my comments as well as the reviewer's reports and your detailed responses (as Word file).

I look forward to reading a new revised version of your manuscript as soon as possible.

Yours sincerely,

Zeljko Durdevic

*** Instructions to submit your revised manuscript ***

- 1) a .docx formatted version of the manuscript text (including Figure legends and tables)
- 2) Separate figure files*
- 3) supplemental information as Expanded View and/or Appendix. Please carefully check the authors guidelines for formatting Expanded view and Appendix figures and tables at <https://www.embopress.org/page/journal/17574684/authorguide#expandedview>
- 4) a letter INCLUDING the reviewer's reports and your detailed responses to their comments (as Word file).
- 5) The paper explained: EMBO Molecular Medicine articles are accompanied by a summary of the articles to emphasize the major findings in the paper and their medical implications for the non-specialist reader. Please provide a draft summary of your article highlighting
 - the medical issue you are addressing,
 - the results obtained and
 - their clinical impact.This may be edited to ensure that readers understand the significance and context of the research. Please refer to any of our published articles for an example.
- 6) For more information: There is space at the end of each article to list relevant web links for further consultation by our readers. Could you identify some relevant ones and provide such information as well? Some examples are patient associations, relevant databases, OMIM/proteins/genes links, author's websites, etc...
- 7) Author contributions: the contribution of every author must be detailed in a separate section.
- 8) EMBO Molecular Medicine now requires a complete author checklist (<https://www.embopress.org/page/journal/17574684/authorguide>) to be submitted with all revised manuscripts. Please use the checklist as guideline for the sort of information we need WITHIN the manuscript. The checklist should only be filled with page numbers where the information can be found. This is particularly important for animal reporting, antibody dilutions (missing) and exact values and n that should be indicated instead of a range.

9) Every published paper now includes a 'Synopsis' to further enhance discoverability. Synopses are displayed on the journal webpage and are freely accessible to all readers. They include a short stand first (maximum of 300 characters, including space) as well as 2-5 one sentence bullet points that summarise the paper. Please write the bullet points to summarise the key NEW findings. They should be designed to be complementary to the abstract - i.e. not repeat the same text. We encourage inclusion of key acronyms and quantitative information (maximum of 30 words / bullet point). Please use the passive voice. Please attach these in a separate file or send them by email, we will incorporate them accordingly.

You are also welcome to suggest a striking image or visual abstract to illustrate your article. If you do please provide a jpeg file 550 px-wide x 300-800px high.

10) A Conflict of Interest statement should be provided in the main text

11) Please note that we now mandate that all corresponding authors list an ORCID digital identifier. This takes <90 seconds to complete. We encourage all authors to supply an ORCID identifier, which will be linked to their name for unambiguous name identification.

Currently, our records indicate that the ORCID for your account is 0000-0002-3876-2209.

Link Not Available

Photos 400-800 DPI

*Additional important information regarding figures and illustrations can be found at

<https://bit.ly/EMBOPressFigurePreparationGuideline>. See also figure legend preparation guidelines:

<https://www.embopress.org/page/journal/17574684/authorguide#figureformat>

***** Reviewer's comments *****

Referee #2 (Comments on Novelty/Model System for Author):

Clearly improved over 1st version (that was already very interesting and sound), by adding mo0re detailed M&M description and some careful rephrasing. Also new dataset added as requested in animal model in support of clinical data.

Referee #2 (Remarks for Author):

The authors made a great effort to answer and discuss all our concerns which is highly appreciated. Especially, addition of data from a well designed mouse trial (using different vaccine doses and metformin treatment schedules) fully support their clinically observed hypothesis. For me things have become much more clear, and the way how now in the revised version methodology and limitation of the study are mentioned may serve as rich basis for future research.

A final minor request is to add a small disclaimer to the results of the animal study (described in Line 243-245). Namely the fairly small sample size (n=5) per group without any biological repeats. Likewise, for the 10e4PFU group results are crystal clear and very much supportive. However, intriguingly, in the 10e5 PFU arm, untreated animals react very consistently to immunisation (very narrow data distribution; all similarly high nAb levels) whereas MET treatment resulted in a large spread of nAb titers - independently of duration of treatment, and geometric mean titers in both treated groups are obviously lower than in untreated animals. For this study arm data are thus less easy to interpret which may deserve a careful mention in the final MS.

***** Reviewer's comments *****

Referee #2 (Comments on Novelty/Model System for Author):

Clearly improved over 1st version (that was already very interesting and sound), by adding more detailed M&M description and some careful rephrasing. Also new dataset added as requested in animal model in support of clinical data.

Referee #2 (Remarks for Author):

The authors made a great effort to answer and discuss all our concerns which is highly appreciated. Especially, addition of data from a well designed mouse trial (using different vaccine doses and metformin treatment schedules) fully support their clinically observed hypothesis. For me things have become much more clear, and the way how now in the revised version methodology and limitation of the study are mentioned may serve as rich basis for future research.

A final minor request is to add a small disclaimer to the results of the animal study (described in Line 243-245). Namely the fairly small sample size ($n=5$) per group without any biological repeats. Likewise, for the 10^4 PFU group results are crystal clear and very much supportive. However, intriguingly, in the 10^5 PFU arm, untreated animals react very consistently to immunisation (very narrow data distribution; all similarly high nAb levels) whereas MET treatment resulted in a large spread of nAb titers - independently of duration of treatment, and geometric mean titers in both treated groups are obviously lower than in untreated animals. For this study arm data are thus less easy to interpret which may deserve a careful mention in the final MS.

We thank Reviewer 2 for the supportive remarks and acknowledge the caveats that were raised regarding our animal study. While the larger spread of neutralizing antibody titers in metformin-treated mice at 10^5 PFU is certainly intriguing, there are a few possible explanations for this observation. Firstly, the immunocompetent C57BL/6 mouse model is incapable of supporting viremic infection when inoculated with a dose between the range of 10^2 to 10^4 PFU which is the dose typically encountered during a mosquito bite. Higher doses, however, can result in breakthrough infections in some animals. Indeed, this phenomenon was also observed by other groups using the same mouse model (Rathore et al, 2021). Alternatively, the vaccine might have been administered into other compartments inadvertently during inoculation. Although the same operator performed the experiment, we cannot exclude this possibility due to the inherent challenges with handling of small animals. However, a lengthy discussion on these various possibilities would distract the readers from the crux of our findings; the importance of metformin in modulating mitochondrial respiration to influence vaccine-induced antibody immunogenicity. We, therefore, respectfully have not added these points into the discussion.

2nd Apr 2024

Dear Dr Low,

We are pleased to inform you that your manuscript is accepted for publication and is now being sent to our publisher to be included in the next available issue of EMBO Molecular Medicine.
